# Localized nuclear reaction breaks boron drug capsules loaded with immune adjuvants for cancer immunotherapy

Yaxin Shi[1,14], Zhibin Guo[1,14], Qiang Fu[2,14], Xinyuan Shen[3], Zhongming Zhang [iD][4], Wenjia Sun[5], Jinqiang Wang[3], Junliang Sun[5], Zizhu Zhang[6], Tong Liu[7], Zhen Gu [iD][3,8,9,10] ✉ & Zhibo Liu [iD][1,11,12,13] ✉

Boron neutron capture therapy (BNCT) was clinically approved in 2020 and exhibits remarkable tumour rejection in preclinical and clinical studies. It is binary radiotherapy that may selectively deposit two deadly high-energy particles ($^4$He and $^7$Li) within a cancer cell. As a radiotherapy induced by localized nuclear reaction, few studies have reported its abscopal anti-tumour effect, which has limited its further clinical applications. Here, we engineer a neutron-activated boron capsule that synergizes BNCT and controlled immune adjuvants release to provoke a potent anti-tumour immune response. This study demonstrates that boron neutron capture nuclear reaction forms considerable defects in boron capsule that augments the drug release. The following single-cell sequencing unveils the fact and mechanism that BNCT heats anti-tumour immunity. In female mice tumour models, BNCT and the controlled drug release triggered by localized nuclear reaction causes nearly complete regression of both primary and distant tumour grafts.

Over 50% of cancer patients benefit from radiotherapy when used alone or in combination with surgery or chemotherapy[1]. Traditional radiotherapy modalities utilize low LET (linear energy transfer) radiation that damages DNA through the indirect effect by producing oxidative radicals (i.e. OH•)[2]. Therefore, oxygen is required in the fixation of DNA damage and hypoxia tumours often resist conventional radiotherapy[3,4]. As a frontier radiotherapy modality that was just approved in clinics,

boron neutron capture therapy (BNCT) induces cell damage through a localized nuclear reaction that emits an alpha particle and a lithium particle[5]. Their LETs are 2-3 magnitudes higher than X-ray and γ-ray, and most of their energy deposits within the radius of a cancer cell (5 to 10 μm)[6]. Its cell-killing process is less sensitive to oxygen and might promote uniform cell killing in tumours despite intermittent normoxic and hypoxic regions[7]. It is worth noting that clinical trials of BNCT have

[1]Beijing National Laboratory for Molecular Sciences, Radiochemistry and Radiation Chemistry Key Laboratory of Fundamental Science, Key Laboratory of Bioorganic Chemistry and Molecular Engineering of Ministry of Education, College of Chemistry and Molecular Engineering, Peking University, Beijing 100871, China. [2]The Centre of Nanoscale Science and Technology and Key Laboratory of Functional Polymer Materials, Institute of Polymer Chemistry, College of Chemistry, Nankai University, Tianjin 300071, China. [3]Key Laboratory of Advanced Drug Delivery Systems of Zhejiang Province, College of Pharmaceutical Sciences, Zhejiang University, Hangzhou 310058, China. [4]Engineering Department, Lancaster University, Lancaster, Lancashire LA1 4YW, UK. [5]College of Chemistry and Molecular Engineering, Beijing National Laboratory for Molecular Sciences, Peking University, Beijing 100871, China. [6]Beijing Nuclear Industry Hospital, Beijing 100045, China. [7]Beijing Capture Tech Co. Ltd, Beijing 102413, China. [8]Jinhua Institute of Zhejiang University, Jinhua 321299, China. [9]Department of General Surgery, Sir Run Run Shaw Hospital, School of Medicine, Zhejiang University, Hangzhou 310016, China. [10]Liangzhu Laboratory, Zhejiang University Medical Center, 311121 Hangzhou, China. [11]Peking-Tsinghua Center for Life Sciences, Peking University, 100871 Beijing, China. [12]Changping Laboratory, 102206 Beijing, China. [13]Key Laboratory of Carcinogenesis and Translational Research (Ministry of Education/ Beijing), NMPA Key Laboratory for Research and Evaluation of Radiopharmaceuticals (National Medical Products Administration), Department of Nuclear Medicine, Peking University Cancer Hospital & Institute, 100142 Beijing, China. [14]These authors contributed equally: Yaxin Shi, Zhibin Guo, Qiang Fu. ✉e-mail: guzhen@zju.edu.cn; zbliu@pku.edu.cn

shown notable efficacy in eliminating locally aggressive tumours, including glioblastoma, melanoma, and recurrent head and neck cancer[8–10]. However, as a local treatment technique, the control of distant metastasis (the main cause of cancer death) by BNCT remains to be explored. Though preliminary studies have observed abscopal effects induced by local BNCT, it is rather occasional[11,12].

90% cancer patients died due to metastasis, while radiotherapy is often prescribed for treating local tumours[13]. The abscopal effect of radiotherapy may lead to regression of metastases, and has been connected to mechanisms involving the immune system[3,14]. Studies have been trying to elucidate how combinational therapy may boost the abscopal effect[15,16], yet their efficacy is often limited because the tumour-induced immune tolerance may hamper the development of radiotherapy-induced immune responses[17]. Combination treatment with immune adjuvants (e.g. imiquimod as a pro-inflammatory agent) is one of the most effective ways to overcome immunosuppression and to improve the immune response[18]. The previous studies indicate that sufficient adjuvants in tumours and killing the tumour cell with activating the tumour immune microenvironment are crucial to success[17], yet it has been challenging to meet the above requirements in clinical practice[19].

In this work, we start by exploring the idea of whether a carborane-based covalent organic framework (B-COF) can be based to develop a boron "capsule" of immune adjuvants for concurrent BNCT and immunotherapy. BNCT occurs when a thermal neutron is captured by a $^{10}B$ atom, resulting in the emission of an alpha particle ($\sim 1.47 \to 1.78$ MeV), a recoiling $^7Li$ nucleus ($\sim 831.6$ keV $\to 1.01$ MeV) and a 478 keV gamma-ray (Fig. 1A)[20]. We find that neutron irradiation could accelerate the release rate of the payload. Many defects have been observed on the frameworks of the boron capsule after neutron irradiation (Fig. 1B). In addition, the results of single-cell sequencing studies demonstrate that BNCT significantly increases the level of total tumour-infiltrating immune cells, facilitates the transformation of immunosuppressed tumours into immunogenic tumours. The sustained release of imiquimod promotes macrophage polarization further increasing the antitumour immune response. Such treatment significantly increases cytokine secretion and the infiltration of functional CD4 + and CD8 + T cells, consequently turning "cold" tumour "hot", showing remarkable growth inhibition to both primary tumour and distant tumours in two types of xenografts-bearing mice.

## Results

### Design and synthesis of B-COF as the boron capsule that loaded with immune adjuvants

To meet the requirement that tumour-localized and long-retention of boron contents and sustainable release of immune adjuvants, μm-scale drug vehicles are preferred instead of nm-scale materials due to their notably longer tumour retention and larger capability for storing small molecule drugs[21]. Featured by large specific surface area and periodic skeletons, covalent organic framework (COF) attracts increasing attention for controlled drug release[22]. The uniform pore size not only improves the loading efficiency of drugs but also facilitates the drug release from the inside to the surface compared to the amorphous polymers with irregular porosity[23,24]. Besides, the intrinsically biological stability of COF is also a key feature that fits the requirement of in vivo studies. Therefore, we proposed that μm-scale B-COF as a boron capsule, which is loaded with a toll-like receptor 7 agonist—imiquimod, for combined BNCT-immunotherapy.

We install carborane as the boron-containing building block (Fig. 1C) for COF due to its high molar boron content and biological inertness[25]. The carborane-linked COF with high crystallinity is synthesized through a Schiff base condensation. Great efforts have been taken in the optimization of reaction conditions, and the synthetic details and material characterization are described in Supplementary Fig. 1-4 and Supplementary Table 1. The crystallinity and unit cell parameters of

B-COF were determined by powder X-ray diffraction (PXRD) analysis (Fig. 1D, red curve and grey curve) and structure simulation (rest in Fig. 1D). Full profile pattern matching (Pawley) refinements were carried out on the experimental PXRD pattern. The peaks of B-COF at $2\theta = 2.06$, 3.53, 4.04 and 16.60° correspond to the (100), (110), (200) and (001) planes, respectively. Simulation by Materials Studio suggests that B-COF crystallizes in the $P3$ space group, and the unit cell parameters are $a = b = 51.2$ Å, $c = 5.3$ Å, $\alpha = \beta = 90°$ and $\gamma = 120°$. We have tried to match the simulated PXRD patterns of different stacking models with the experimental results (Supplementary Fig. 5 and Supplementary Table 2–4), and found that the COF is AA-stacking. The refinement results produced good agreement factors ($\omega R_p$, the weighted profile $R$-factor = 4.77% and $R_p$ = 3.50% for B-COF) (Fig. 1D).

The transmission electron microscope (TEM) of B-COF shows that the diameter is approximately 1 μm and is granular crystallites affording uniform morphology (Fig. 1E). The hexagonal pores can be visualized (Fig. 1F and 1G) and the average lattice distance is 4.8 nm (Fig. 1H). The elemental mapping showed the geometrical and compositional distributions of C, B, and N in B-COF (C as red, B as blue, N as green, Fig. 1I-K and Supplementary Fig. 6). Elemental analysis suggests that the boron content in B-COF was 20.55 wt%, which coordinates with the calculation about the elemental composition in each structural unit (Supplementary Table 5). The Fourier transform infrared spectra (FTIR) showed the appearance of the peak of -C = N- at 1600 cm$^{-1}$ in B-COF along with the disappearance of the aldehydic -C-H (2923 and 2857 cm$^{-1}$) and -C = O (1696 cm$^{-1}$) of B-CHO and the -N-H (3436 and 3352 cm$^{-1}$) stretching vibrations of TAPB (Fig. 1L), indicating the successful formation of imine bond from the condensation reaction.

Imiquimod-loaded B-COF was prepared through vigorous stirring in ethanol to afford uniform morphology (Supplementary Fig. 7). The obtained PXRD peaks of imiquimod-loaded B-COF are consistent with those of B-COF, suggesting the structural integrity after loading (Supplementary Fig. 8). The Brunauer-Emmett-Teller (BET) surface area of B-COF is 673 m$^2$ g$^{-1}$, with a total pore volume of 0.68 cm$^3$ g$^{-1}$ (pore size, 2.95 nm), which were reduced to 275 m$^2$ g$^{-1}$ and 0.31 cm$^3$ g$^{-1}$, respectively (Fig. 1M and Supplementary Fig. 9) after drug loading. Thermogravimetric analysis (TGA) shows that the decomposition temperature is 503 °C for B-COF and the weight loss of imiquimod-loaded B-COF is 53% (Fig. 1N), the calculated imiquimod loading efficiency by UPLC-MS was 37.1%. The electrostatic surface potential maps (ESPs) revealed that imiquimod is mainly localized near benzene instead of carborane in B-COF, the binding energy between imiquimod and benzene (−3.5 kcal/mol) is higher than that binds to carborane (−0.7 kcal/mol) (Supplementary Fig. 10). Imiquimod-loaded B-COF were coated with DSPE-PEG for better physiological stability and dispersibility. TEM shows that the obtained boron microspheres were well-dispersed in aqueous solution and exhibited spherical shape with homogeneous size (Supplementary Fig. 11). The dynamic light scattering (DLS) study shows that the hydrodynamic diameter is 1002.1 ± 121.6 nm and the zeta potential is −46.7 ± 4.0 mV (Fig. 1O). The biological stability of boron microspheres is excellent in both phosphate-buffered saline (PBS) and serum (Supplementary Fig. 12), which can be readily used for further tests in vitro and in vivo.

### Drug release promoted by boron neutron capture nuclear reaction

Tumour concentration of immune adjuvants and killing tumour cell with the activation of tumour immune microenvironment are the keys for upregulating tumour immune response of radiotherapy[17]. We have been curious that whether boron neutron capture nuclear reaction, after which a boron atom "disappears" in situ to create new channels within frameworks for small molecule transportation, would increase the drug release right after the neutron irradiation. It may not be practical, as in theory only several $^{10}B(n,\alpha)^7Li$ nuclear reactions would happen in each B-COF. The limited defects may not result in a notable

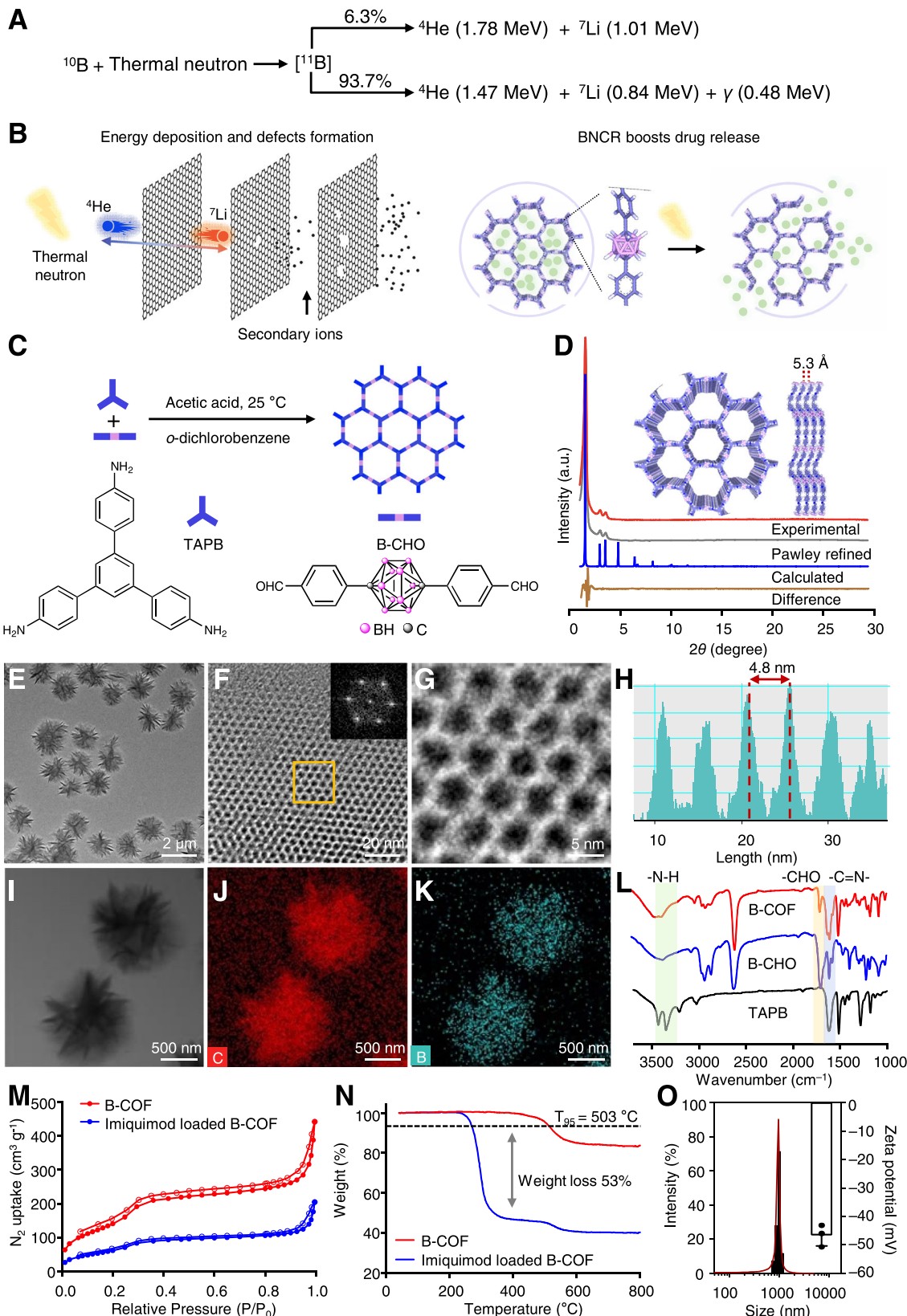

acceleration of drug release. Interestingly, neutron irradiation–drug release assay (Fig. 2A), determined by UPLC-MS, showed a significantly faster release of imiquimod from the boron microspheres after neutron irradiation (Fig. 2B). We wondered whether there is any radiolysis of imiquimod due to the nuclear reaction, and we did not observe any fragment or degradation from imiquimod according to UPLC-MS

(Supplementary Fig. 13A). In addition, imiquimod was released in a sustained manner over one month in vitro (Supplementary Fig. 13B), showing their long-term efficacy in vivo.

We have been wondering the possible mechanism of this neutron irradiation–triggered acceleration of drug release. X-ray photoelectron spectroscopy (XPS) and PXRD (Supplementary Fig. 13C

**Fig. 1 | Boron capsule is assembled as a PEGylated drug-loaded carborane-derived covalent organic framework. A** Boron neutron capture reaction (BNCR) occurs when $^{10}B$ is irradiated with thermal neutrons, yielding fatal and high-energy alpha particles ($^{4}He$) and lithium-7 nuclei ($^{7}Li$). **B** High-energy nuclei from BNCR break boron capsule through inelastic collisions. The resulting secondary ions develop more defects in the framework and therefore construct additional channels for faster drug release. **C** Schematic representation of the synthesis of carborane-derived covalent organic framework (B-COF) by the condensation of 1,3,5-tris(4-aminophenyl)-benzene (TAPB) and p-carborane-1,10-phenyl-dialdehyde (B-CHO) under an optimized condition. **D** Powder X-ray diffraction (PXRD) patterns of the experimentally observed (red curve), Pawley refined (grey curve) B-COF and an eclipsed AA stacking mode (blue curve) (their difference is shown in the green curve). The inset shows reconstructed structure of AA stacking layered framework of B-COF at top view and side view, respectively. **E** Large-scale transmission electron microscopy (TEM) of B-COF with uniform morphology. **F, G** TEM image of B-COF showing hexagonal pores. Inset of panel **G** is the fast Fourier transform (FFT) from the orange square marked area. **H** Lattice fringe distance measurement of the marked area in panel **H** showing the average lattice distance of B-COF. **I, J** TEM (**I**) and the energy dispersive spectroscopy (EDS) elemental mapping images of carbon (**J**) or boron(**K**) of B-COF. **L**, Fourier transform infrared spectra (FTIR) of B-COF and the corresponding building blocks ($n = 3$ samples), respectively. **M** Nitrogen adsorption (solid dots)-desorption (hollow dots) isotherms at 77 K of B-COF and drug-loaded B-COF ($n = 3$ samples). **N** Thermogravimetric analysis (TGA) curves of B-COF and drug-loaded B-COF ($n = 3$ samples). **O** Hydrodynamic size and zeta potential ($n = 3$ samples) of the boron capsule (the PEGylated drug-loaded B-COF). The experiments for **E–K** were repeated three times independently ($n = 3$ samples) with similar results. Data are presented as mean ± SD (**O**). Source data are provided as a Source Data file.

and 14) showed no significant difference before and after neutron irradiation, indicating that there is no major disintegration in the framework. An unanticipated finding is from TEM browsing B-COF under different field of views (Fig. 2C and 2D). We found that there are some defects randomly distributed in B-COF after neutron irradiation. It is worth noting that these defects appear in clusters, but one cluster of defects is often distant from others. These defects can be large—the size is up to several micrometers, which is of great importance for rapid drug release[26,27]. The vacancies are generated due to the sputtering of atoms, and their size and number are highly correlated to the energy deposit of $^{4}He$ and $^{7}Li$ in B-COF. Simulation by Monte Carlo calculation (Fig. 2E) indicates that they can deposit intense energy ($^{7}Li$, 2.59 keV or 3.52 keV; $^{4}He$, 1.31 keV or 1.88 keV; depends on the simulation models, respectively) through inelastic collisions to the atoms in the framework (Fig. 2F and G, Supplementary Table 6–9), which were 2-3 magnitudes higher than the energy of chemical bond (<10 eV)[28]. It is therefore suggested that the daughter nuclei from boron neutron capture reaction and their secondary ions can develop many defects in B-COF, thus constructing new channels that accelerating drug release.

The molecular dynamics (MD) simulation helps further understand the nature of vacancy generation process[29]. As shown in Fig. 2H, a $^{7}Li$ nucleus (red dot) generated from boron neutron capture reactions is of high energy and can knock out many atoms (grey dots), which as the secondary ions may continue striking other atoms in different layers, giving additional defects of different sizes (due to different energy of the secondary ions) in the materials. This simulation suggests that the defects generation by boron neutron capture reaction is highly efficient, which coordinates our observation, unveiling a method for engineering defects for boron-containing materials. In addition, this strategy matches well with our boron microspheres, achieving boron neutron capture therapy-induced "rupture" to release adjuvants.

**Pyroptotic-like cell death with neutron irradiation**
The cytotoxicity of boron capsule has been evaluated in B16F10 murine melanoma cells and MC38 murine colorectal cancer cells. As desired, the boron capsule is biologically safe, the cell viabilities are >90% for the groups treated with up to 1 mg/mL of the boron capsule (Fig. 3A). Sufficient boron contents (>20 μg per $10^{7}$ cells) in cancer cells are the key for BNCT, yet may be challenging for μm-scale boron capsule[10,30]. To investigate whether the boron capsule could be efficiently endocytosed by tumour cells, we have tried to track the FITC-labeled boron capsule in B16F10 and MC38 cells through confocal microscopy. As shown in Fig. 3B, rapid cell uptake was observed in the cytoplasm within 2 h, and significantly more boron capsules were found in both cell lines at 24 h post-incubation. The average number of boron capsules per cell increased from 8.9 ± 3.6 to 45.7 ± 9.9 for B16F10 tumour cells and 8.4 ± 4.0 to 37.2 ± 8.7 for MC38 tumour cells from 2 h to 24 h, respectively (Fig. 3C). In contrast, the average number

of boron capsules per cell after 24 h of incubation was 9.2 ± 5.0 for primary mouse bone-marrow-derived macrophages (BMDMs) compared with that for B16F10 tumour cells (Supplementary Fig. 15A-B), suggesting that boron capsules may be endocytosed by tumour cells more than immune cells. Coordinating with the increasing endocytosis of boron capsule, the boron contents per $10^{7}$ cells increased from 40.7 ± 8.5 to 212.8 ± 4.8 μg for B16F10 tumour cells, and from 37.3 ± 4.8 to 172.6 ± 9.6 μg for MC38 tumour cells from 2 h to 24 h, respectively (Fig. 3D). The boron contents in B16F10 and MC38 cells at 24 h is 5.2-fold and 4.6-fold to the boron contents at 2 hours, respectively.

Subsequently, the cell treatment efficacy of BNCT with boron capsule was determined by CCK8 assay and flow cytometry. The cells were treated by boron capsule, PEGylated B-COF (PEG-B-COF), imiquimod or PBS with (+) or without neutron irradiation. Using this assay, we found that boron capsule+neutron and PEG-B-COF + neutron treatment exhibited notable growth inhibition to both B16F10 and MC38 cancer cells, which other treatments showed minimal cytotoxicity (Fig. 3E). In contrast, consistent with the low cellular uptake of boron capsule, a moderate cell-killing occurred in the BMDMs (Supplementary Fig. 15C), which may indicate that only a small portion of immune cells were directly impacted by BNCT compared to tumour cells. This result coordinates with the colony formation assays shown in Supplementary Fig. 16. Interestingly, flow cytometry assay shows that BNCT with boron capsule rendered 64.2% and 55.9% of B16F10 and MC38 cells, respectively, exhibit double-positive (propidium iodide +/ Annexin-V +, Fig. 3F and 3G), which indicates notable membrane damage and is often a hallmark for pyroptotic cell death[31,32]. This suggests that BNCT with boron capsule may be pro-inflammatory, which is the key to "heat" tumour immunity.

**Inhibition of tumour growth by bystander effect**
This encouraged us to investigate the feasibility of using BNCT with boron capsule to treat the primary tumour and to provoke tumour immunity. We examined the distribution of intratumourally injected boron capsule in the tumour-bearing mice by PET imaging. For this, [$^{89}Zr$]boron capsule was prepared as previously described and injected in an intratumoural manner[32]. Nearly all [$^{89}Zr$]boron capsule remain in the tumour within 24 h, with little leakage to the circulation system or major organs (Fig. 4A). The TEM characterization shows that boron capsules could be taken up into tumour cells in mice and were distributed in the cell cytosol (Fig. 4B). The TEM analyses allowed to capture each step of boron capsule trafficking, including those becoming completely exposed to the cell cytosol (Fig. 4B, Field 1 and Field 2) or residing in the intact (Fig. 4B, Field 3). Of note, intracellular boron capsules keep their original morphology, which shows the excellent in vivo stability of boron capsule.

Next, we tested whether the following BNCT could reject the tumour growths in mice. C57BL/6 mice engrafted subcutaneously with the B16F10 cancer cells were treated with boron capsule or an equal amount of PEG-B-COF at day 6 followed by sequential neutron

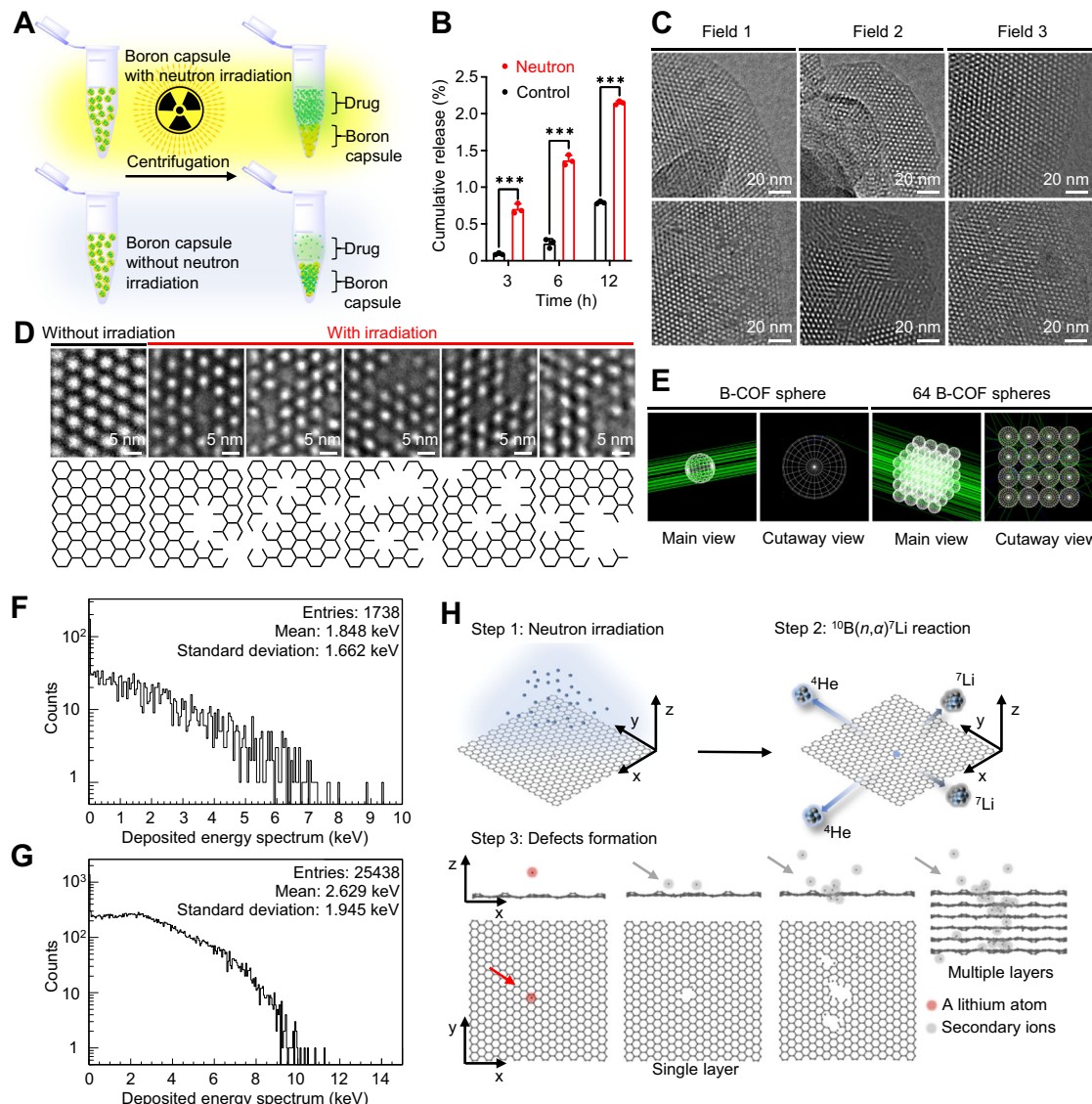

**Fig. 2 | Defects induced by neutron capture nuclear reaction accelerate the drug release from the boron capsule. A** Workflow of assaying drug release from neutron-activated boron. Boron capsule in phosphate-buffered saline (PBS, pH = 7.4) was exposed to thermal neutron irradiation for 30 min with a thermal neutron flux of $1.9 \times 10^{9}/(cm^2 \cdot s)$. The released drug (i.e. imiquimod) was isolated by centrifuge and then determined by ultra-performance liquid chromatography-mass spectrum (UPLC-MS). **B** In vitro drug release profile of boron capsule in PBS at 37 °C. Data are shown as mean ± SD. ($n = 3$ independent experiments, two-tailed unpaired Student's $t$-test, $P$ value <0.0001). **C** TEM images of B-COF before (top) and after (bottom) thermal neutron irradiation ($n = 3$ samples). Notable defects were observed after irradiation. **D** Representative enlarged TEM images (top) and structure models (bottom) of B-COF with or without irradiation to give more details of defects ($n = 3$ samples). **E** Schematic models of Monte Carlo simulation for a 1-μm B-COF sphere and 64 1-μm B-COF spheres. **F, G** The energy spectra of particle deposition investigated by Monte Carlo simulation in 1-μm B-COF sphere and 64 1-μm B-COF spheres, respectively. **H** Schematic illustration of defect formations investigated by molecular dynamics (MD) simulation with a simplified graphene model. Step 1, A high flux thermal neutron reacted with carborane though $^{10}B(n,\alpha)$ $^{7}Li$ reaction; Step 2, The fission-generated high-energy ions (i.e. alpha particles and lithium-7 nuclei) deposits energy through inelastic collisions to atoms in B-COF; Step 3, The high-energy ions break B-COF framework and produce many secondary ions, which further promote the development of defects.

irradiation at day 7, respectively (Supplementary Fig. 17). The dose composition of normal and tumour tissues based on the average boron concentration in tissues during neutron irradiation calculated by the Simulation Environment for Radiotherapy Applications (SERA) system is shown in Fig. 3. The growth of B16F10 tumours is rather aggressive. In PBS-treated control mice, the tumour volumes increased expectedly by 10-fold in 10 days (Fig. 4D, Supplementary Fig. 18A, Supplementary Table 10). Strikingly, compared with the control group, a pronounced growth delay over 40 days was observed with boron capsule+neutron irradiation–treated mice. In contrast, mice treated with boron capsule or neutron irradiation alone behaved similarly to the PBS-treated mice and showed normal and aggressive tumour growth. It is worth noting that mice treated with PEG-B-COF also exhibited remarkable inhibition

to tumour growth, yet the efficacy is heterogeneous as rapid recurrences were observed in two mice at a late time point. This suggests that the neutron-induced release of an immune adjuvant may be essential to achieve comprehensive tumour treatment. Of note, no obvious loss of body weight was observed in all groups (Fig. 4E) and the results of serum biochemical test, routine blood analysis (Supplementary Fig. 19) and Hematoxylin-Eosin (HE) staining of major organs (Supplementary Fig. 20) showed negligible systemic toxicity of boron capsule and BNCT. These indicate that BNCT with boron capsule is well tolerated in mice, highlighting the biological safety of this therapeutic strategy. A similar tumour regression effect was observed with boron capsule+neutron irradiation treatments in mice engrafted with another mouse tumour cell line MC38 (Fig. 4F-G, Supplementary Fig. 18B).

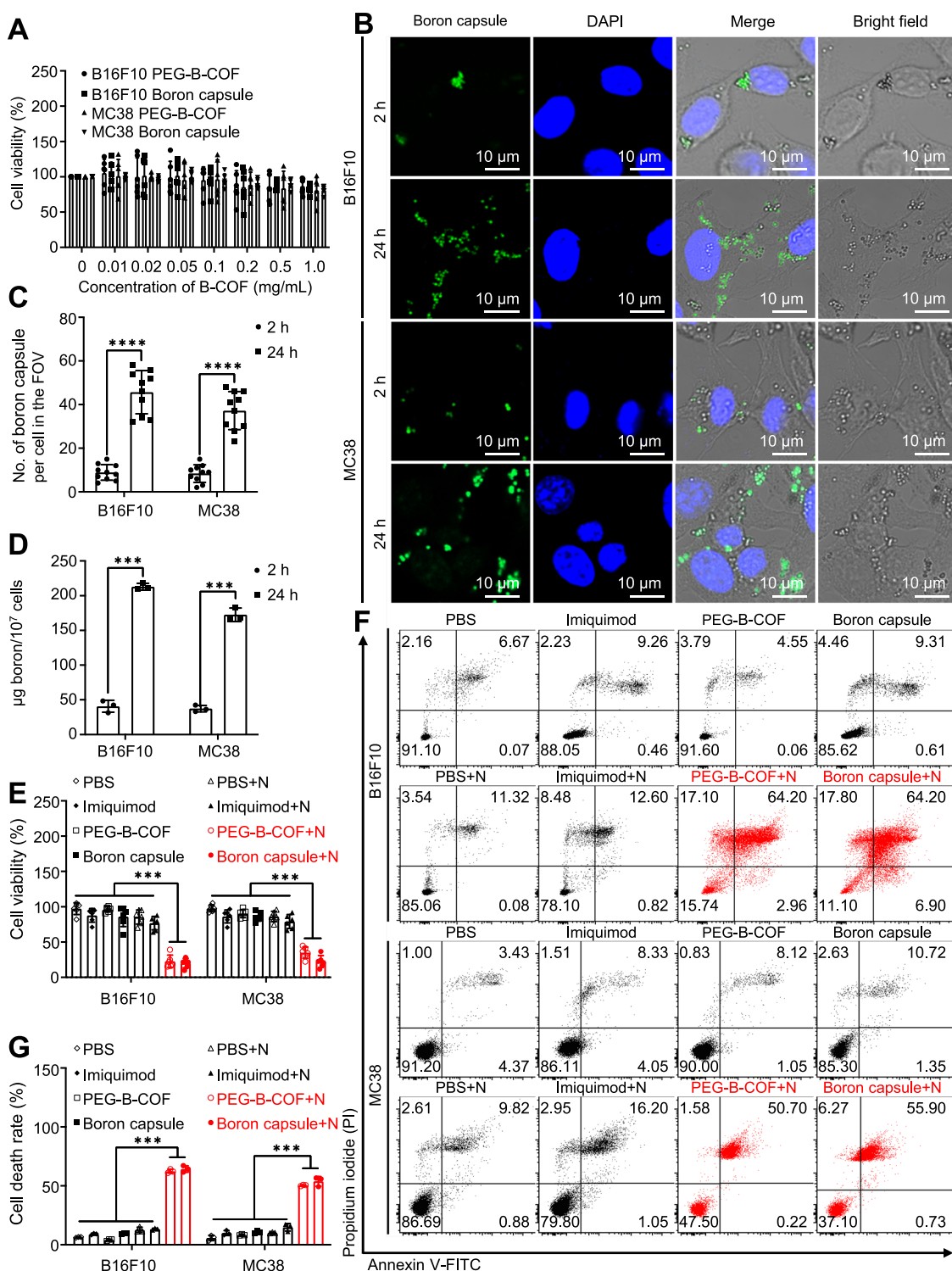

**Fig. 3 | Cell uptake of boron capsule is notable and induces efficient cell death after neutron irradiation. A** Cell viability assays of boron capsule and PEG-B-COF in B16F10 and MC38 cancer cells ($n = 6$ independent experiments, $P$ value < 0.0001), respectively. **B** Cell uptake of boron capsule is time-dependent. Representative confocal images of B16F10 and MC38 cells are shown after being treated by FITC fluorophore-conjugated boron capsule (1 mg/mL) for 2 hours and 24 hours, respectively. Green fluorescence clearly shows the cyto-distribution of boron capsules. **C** Number of boron capsule per cell in the giving field of view (FOV) by confocal microscopy ($n = 10$ fields of view). **D** Boron contents in cancer cells ($n = 3$ independent experiments, $P$ value <0.0001). **E** Efficacy of BNCT on cell viability of B16F10 and MC38 cells ($n = 6$ independent experiments, $P$ value <0.0001). The cells were treated by boron capsule, B-COF, imiquimod or PBS with (+N) or without neutron irradiation. **F** B16F10 or MC38 were treated as indicated. Flow-cytometry plots of PI- and annexin V−FITC-stained cells are shown. **G** The cell death rate was calculated as Annexin V−FITC + /PI + cells and Annexin V−FITC + /PI − cells. ($n = 3$ independent experiments, $P$ value <0.0001). **C, D,** Two-tailed unpaired Student's $t$-test (***$P$ < 0.001, ****$P$ < 0.0001). **E, G,** One-way analysis of variance (ANOVA) followed by Tukey's honest significant difference (HSD) post hoc test (***$P$ < 0.001). Data are presented as mean ± SD (**A, C, D, E, G**). Source data are provided as a Source Data file.

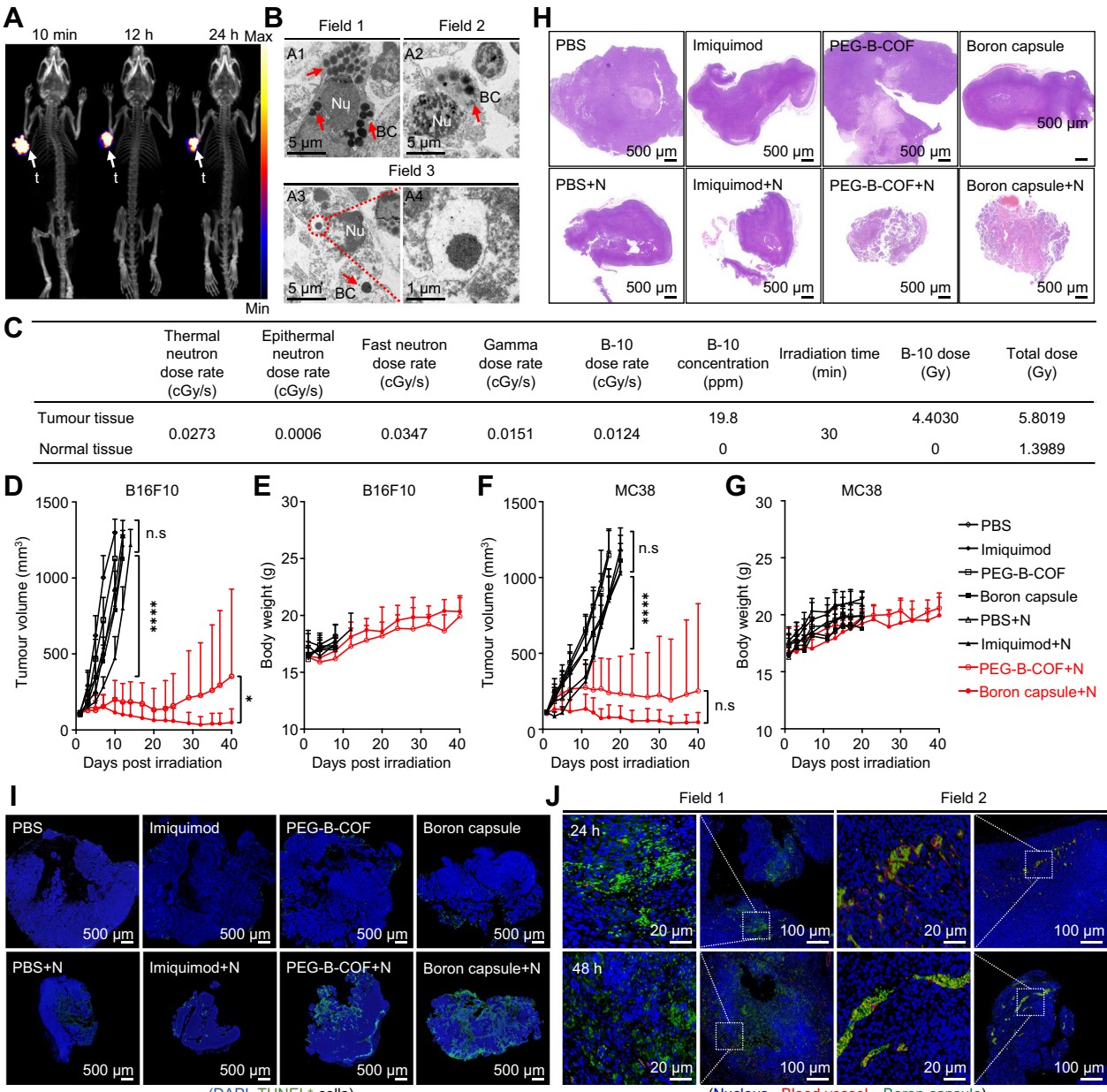

**Fig. 4 | Neutron capture therapy with boron capsule induces tumour regression in mice. A** Dynamic positron emission tomography-computed tomography (PET-CT) 3D projection images of B16F10 tumour-bearing mice at the indicated time points after intratumoural injection of [⁸⁹Zr]boron capsule ($n = 3$ mice).
**B** Representative TEM images show the excessive cyto-distribution of boron capsule in B16F10 tumour tissue. Nu, nucleus; BC, boron capsule ($n = 3$ mice). **C** Dose composition of normal and tumour tissues during neutron irradiation. Dosimetry evaluation of irradiation was based on mean boron concentrations of tumour and normal tissues 24 h post-administration ($n = 3$ mice). **D–G** Assays of tumour growth in mice treated with neutron capsule+neutron irradiation. The mice were treated by boron capsule, PEG-B-COF, imiquimod or PBS with (+N) or without neutron irradiation ($n = 6$ mice). Data shown as mean ± SD. (two-tailed unpaired Student's *t*-test, (****$P < 0.0001$). Treatments began when the tumour volume reached 100 ± 25

mm³. **D, F** Average tumour volumes in the mice inoculated with B16F10 xenografts (**D**) and MC38 xenografts (**F**), respectively. **E, G** Bodyweight of each group of mice. **H** Hematoxylin-Eosin (HE) analysis of B16F10 tumour slices at 7 days after the indicated treatment. Remarkable shrinkage of the nuclei is observed in the BNCT treatment groups. **I** DNA degradation assay in tumours at 1 day after the indicated treatment. Representative confocal images of tumour slices are shown. DNA fragments were assayed with terminal deoxynucleotidyl transferase dUTP nick end labeling (TUNEL, green fluorescence) and the nuclei were stained with DAPI (blue). **J** Representative immunofluorescence staining images of FITC-conjugated boron capsule in B16F10 tumour-bearing mice. The experiments for **H**, **I** and **J** were repeated three times independently ($n = 3$ mice) with similar results. Source data are provided as a Source Data file.

As the "effective killing radius" of BNCT is highly localized (<10 μm, about the diameter of a cell), heterogeneous distribution of boron contents may result in restricted treatment of cancer cells in the tumour[33]. Interestingly, HE analysis at 7 days post-BNCT suggests that the remarkable shrinkage of the nucleus is observed through the entire B16F10 tumours (Fig. 4H). This is contradictory to aforementioned hypothesis. Representative confocal images of tumour slices showing that the detection of DNA fragments is relatively heterogeneous (TUNEL staining, shown as green fluorescence, Fig. 4I). This is not unexpected, as the confocal images of B16F10 tumour slice show that the distribution of boron capsule is isolated at 24 and 48 hours post intratumoural injection (Fig. 4J).

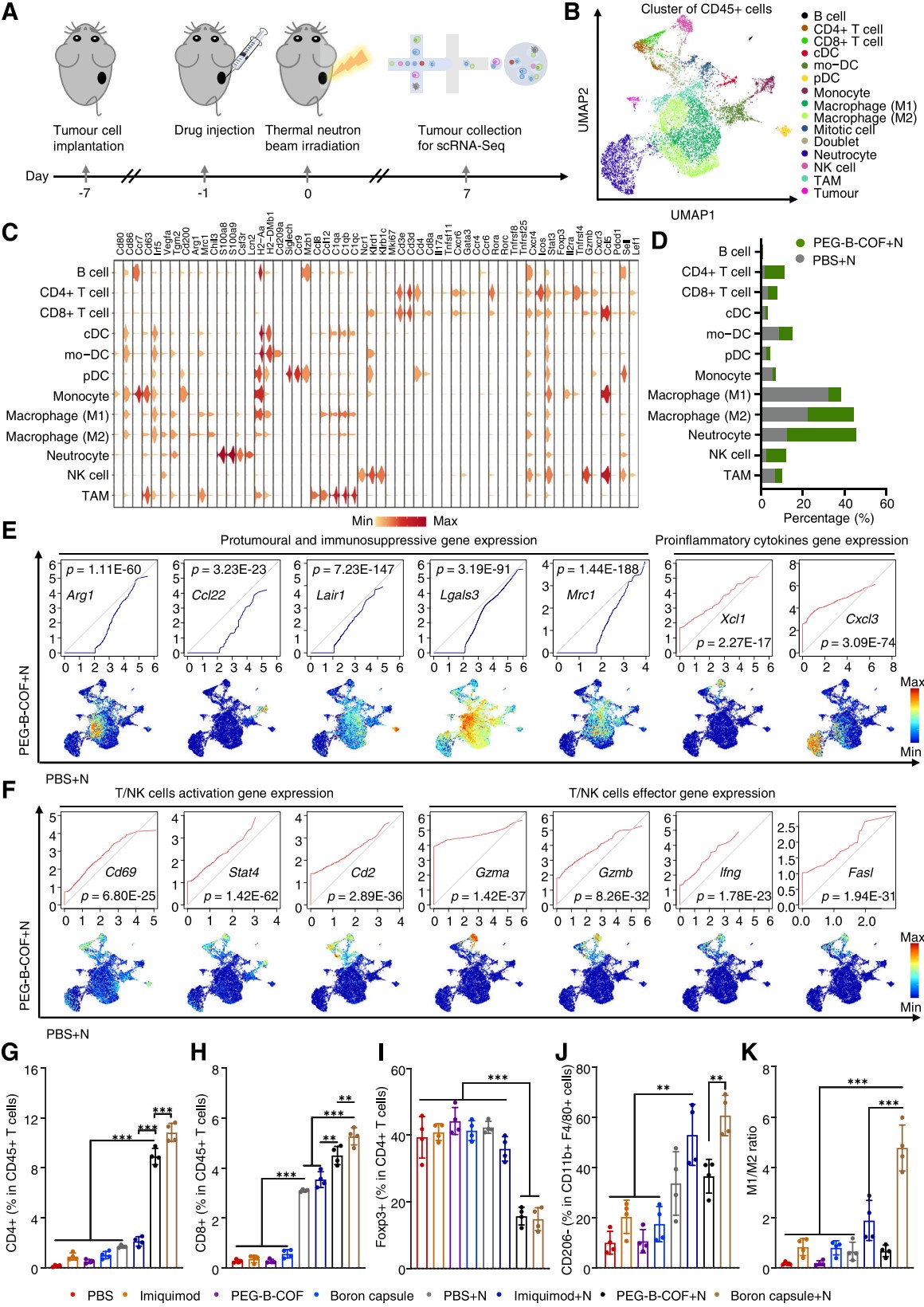

However, the limited ratio of tumour cells in DNA damage could not directly account for the dramatic regression observed with the entire tumour. Besides, this bystander effect can be promoted by immune adjuvants, as BNCT with boron-capsules shows significantly better treatment efficacy than that of BNCT with PEG-B-COF.

## BNCT remodels tumour immune microenvironment

To obtain a full picture of BNCT-triggered immunological remodelling in the tumour microenvironment, the CD45 + immune cells from B16F10 tumours treated by PBS + neutron and PEG-B-COF + neutron were subjected to single-cell RNA sequencing (scRNA-Seq) analyses (Fig. 5A). The total immune cells from different groups were

**Fig. 5 | Single-cell RNA sequencing and FACS analysis shows that BNCT triggers antitumour immunity in primary tumours. A** Schematic illustration of the enrichment of B16F10 tumour-infiltrating CD45 + immune cells. **B** Uniform manifold approximation and projection (UMAP) plots of tumour-infiltrating CD45 + immune cells of B16F10 tumours treated by neutron irradiation and BNCT, and the relative frequencies of different clusters. Conventional dendritic cell, cDC. Monocyte-derived dendritic cell, mo-DC. Plasmacytoid dendritic cell, pDC. Natural killer, NK. Tumour-associated macrophage, TAM. Each dot corresponds to one single cell, colored according to cell cluster. **C** Violin plots show the distribution of expression for marker genes across all cell type clusters. **D** Relative tumour-infiltrating CD45 + immune cells frequencies of different clusters with and without BNCT. **E, F** Expression levels of protumoural and immunosuppressive genes, proinflammatory chemokine (**E**) and T and/or natural killer cell activation or effector (**F**) genes in immune cells. Paired quantile-quantile (Q–Q) plots were used to compare the gene-expression levels in CD45 + immune cells between tumours treated with neutron or BNCT. *P* values were calculated using a two-sided Wilcoxon rank-sum test. **G–K** Quantification of infiltrating CD4 + (**G**), CD8 + (**H**), Foxp3 + (**I**), M1 macrophage (**J**) and M1/M2 ratio (**K**) from the B16F10 tumours. Data is shown as mean ± SD. (*n* = 4 mice). One-way analysis of variance (ANOVA) followed by Tukey's HSD post hoc test. \**P* < 0.05, \*\**P* < 0.01, \*\*\**P* < 0.001. Gating/ strategies of **G–K** is provided in Supplementary Fig. 23-24. Source data are provided as a Source Data file.

sequenced and clustered on the uniform manifold approximation and projection (UMAP) plots (Fig. 5B). These subsets were identified or classified by the differential expression of signature genes (Fig. 5C, Supplementary Fig. 21 and 22). Compared with PBS-treated tumours, scRNA-Seq analysis suggested that PEG-B-COF + neutron-treated tumours had increased populations of CD4 + , CD8 + and NK (natural killer) cells but decreased percentage of myeloid cells (Fig. 5D). In PEG-B-COF + neutron-treated tumours, expression of pro-tumoural and immunosuppressive genes (*Arg1, Ccl22, Lair1, Lgals3* and *Mrc1*) were downregulated, while pro-inflammatory chemokine genes (*Xcl1* and *Cxcl3*), T/NK cell activation genes (*Cd69, Stat4* and *Cd2*) and T/NK cell effector genes (*Gzma, Gzmb, Ifng* and *Fasl*) were upregulated (Figs. 5E and 5F). FACS analysis suggest that both PEG-B-COF + neutron-treated tumours and boron capsule+neutron-treated tumours, had increased populations of CD3 + , CD4 + and CD8 + (Fig. 5G-H, Supplementary Fig. 25A, Supplementary Table 11-12), and decreased populations of Treg cells (Fig. 5I) compared to other groups. Interestingly, M1 macrophages, which often enhance the proinflammatory effects and facilitate immune response[34], are significantly higher in tumours treated by boron capsule+neutron than those of other groups (Fig. 5J). While M2 macrophages, which often enhance cancer cell metastasis and represent poor prognosis[35], decreased significantly in every group of tumours treated with imiquimod (Supplementary Fig. 25B). Consequently, the combination of BNCT and local release of imiquimod gives the highest M1/M2 ratio over other groups (Fig. 5K). Furthermore, a significantly higher degree of CRT exposure and HMGB1 release, which are the biomarkers commonly used in the characterization of immunogenic cell death (ICD), was observed in both PEG-B-COF + N and boron capsule+N groups (Supplementary Fig. 26), indicating ICD has been successfully induced by the BNCT. The above results unveil the anti-tumour immunological effect of BNCT. We also found that the immune response can be further activated when combines with FDA-approved adjuvants.

### Treating distant and metastatic tumours by abscopal antitumour effect

The bilateral tumour-bearing animal model was established by subcutaneous injection of B16F10 (or MC38) cancer cells into the right and left flank regions of C57BL/6 mice (Fig. 6A). The tumour on the right was designated as the primary tumour and received local neutron irradiation, while the tumour on the left was designated as a distal tumour with no direct irradiation (Fig. 6B). Serum cytokines including tumour necrosis factor-α (TNF-α), interleukin 12 (IL-12p70) and interleukin-6 (IL-6) have been assayed by the enzyme-linked immunosorbent assay (ELISA) at 7 days after the BNCT (Fig. 6C and Supplementary Table 13).

The cytokine levels in boron capsule+neutron–treated tumours are significantly higher than those of other groups, including the PEG-B-COF + neutron–treated tumours. This result indicates that a systematic anti-tumour immune response emerges, as IL-12 plays important role in activating natural killer cells[36], and IL-6 and TNF-α are critical in the activation of tumour immunity[37]. Therefore, compared to other treatment groups, populations of CD3 + , CD4 + and CD8 +

increased while Treg decreased significantly in the distant tumours (Fig. 6D–G and Supplementary Table 14).

We further tested the treatment efficacy in the distant tumour on B16F10 and MC38-xenografts-bearing mice. In comparison, upon treatment with thermal neutron irradiation, a remarkable inhibition on tumour growth over 40 days was seen with boron capsule +neutron irradiation–treated mice, while mice treated with BNCT alone showed restrained but to a significantly less extent tumour growth (Fig. 6H and Supplementary Fig. 27A). Besides, boron capsule+neutron irradiation–treated mice survived much longer than the others (Fig. 6I). Similarly, obvious distant-tumour regression was observed with boron capsule+neutron irradiation treatments in mice engrafted with MC38 (Fig. 6J-K, Supplementary Fig. 27B and Supplementary Table 15). We have also tested the efficacy of BNCT with boron capsules in a breast cancer model with lung metastases (Supplementary Fig. 28A). As shown in Fig. 6L & M, the degree of inhibition of lung metastasis was significantly higher in the PEG-B-COF + N group compared with the other groups. In addition, BNCT alone showed weaker inhibition of pulmonary metastases than the boron capsule + N group, suggesting that BNCT alone is difficult to elicit a potent immune response. Taken together, these results suggested that BNCT induces immunogenic cell death and exhibited abscopal effect could be not sufficient to stimulate effective antitumour immunity.

## Discussion

In summary, we present a boron capsule with uniform nanopores for anchoring hydrophobic drugs. Application of this stable and biocompatible delivery system enables sustained drug release and enriched adequate boron to cancer cells for BNCT. As a local treatment technique, the abscopal effects of BNCT remain to be explored. Combinational therapy has shown efficacy to boost the abscopal effect both in preclinical and clinical studies. Of note, the boron neutron capture reaction can generate atomic defects or break the skeletons of two-dimensional materials, readily forming channels for drug transportation, thus empowers BNCT as a potential function for controlled drug release. This application enabled us to treat tumours concurrently with BNCT and adjuvants, which uncovers an immunotherapy potential of BNCT. We further found that BNCT-induced tumour regression could be mediated by T cells that show increased infiltration into the tumours. BNCT not only initiates effective antitumour immunity on its own but also can synergize with the controlled release of immune adjuvants to activate a systematic antitumour immune response. This finding highlights the concept that neutron-activated boron capsule holds the promise to potentiate abscopal antitumour effect, that may be clinically practical to treat metastases through localized radiotherapy.

## Methods

All animal experiments were performed according to the Animal Protection Guidelines of Peking University, China. All animal care and experimental procedure were performed by following the animal protocols (CCME-LiuZB-2) approved by the ethics committee of Peking University.

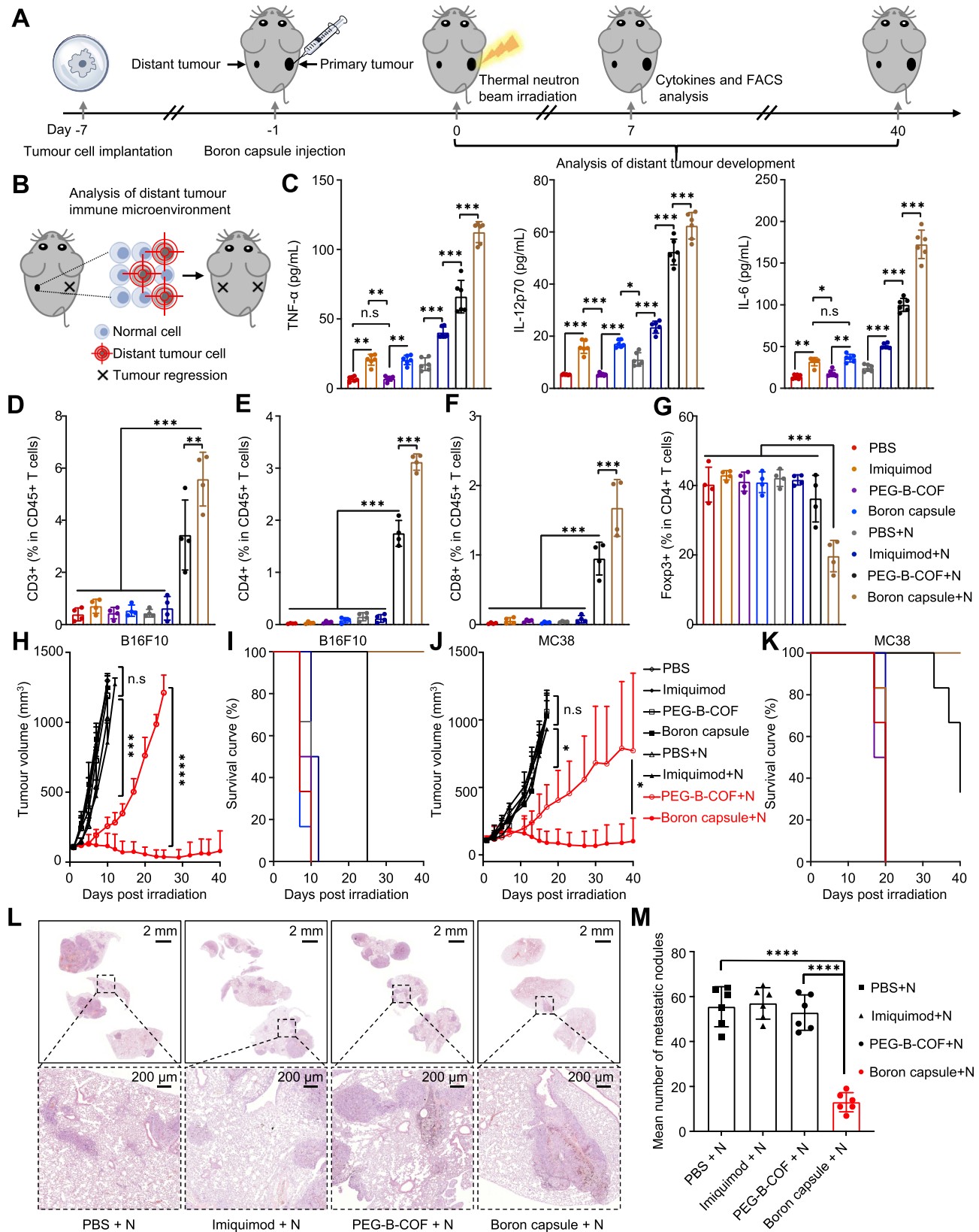

## Reagents

All chemical reagents, purchased from Meryer Chemical (China), Zhengzhou Alfa (China), J&K (China), Energy Chemical (China), Xi'an Ruixi Biological Technology Co., Ltd (China) and Macrocyclics, Inc. Cell counting kit-8 (CCK-8) was purchased from Biyuntian

Biotechnology Institute. Annexin V-FITC/PI Apoptosis Detection Kit was purchased from Yeasen Biotechnology (Shanghai) Co., Ltd. $^{89}$Zr was produced on an onsite cyclotron by the $^{89}$Y($p,n$)$^{89}$Zr reaction and purified to yield $^{89}$Zr(C$_2$O$_4$)$_2$ solution. The cell Apoptosis 7-AAD Detection Kit was purchased from KeyGen Biotech Co., Ltd (Nanjing,

**Fig. 6 | BNCT with immunological adjuvant–loaded boron capsule induces systemic antitumour immunity leading to the inhibition of distal and metastatic tumours. A** Treatment scheme. **B** Schematic illustration of the analysis of distant tumour immune microenvironment. **C** Enzyme-linked immunosorbent assay (ELISA) of TNF-α, IL-12p70 and IL-6 in the serum of mice at 7 days after the indicated treatments (*n* = 6 samples). **D**–**G** Quantification of infiltrating CD3 + (**D**), CD4 + (**E**), CD8 + (**F**), Foxp3 + (**G**) immune cell from the distant B16F10 tumours (*n* = 4 mice). **H**–**K** Assays of distant tumour growth in mice treated with neutron capsule+neutron irradiation. The mice were treated by boron capsule, PEG-B-COF, imiquimod or PBS with (+) or without neutron irradiation (*n* = 6 mice). Data shown as mean ± SD. (two-tailed unpaired Student's *t*-test, ***$P < 0.001$). **H, J** Average distant tumour volumes in the mice inoculated with B16F10 xenografts (**H**) and MC38 xenografts (**J**). **I, K** Survival curves. **L** Representative lung H&E staining for each treatment group. **M** The number of metastatic nodules in different groups (*n* = 6 mice). (**C, D, E, F, G, M**), data are shown as mean ± SD, one-way analysis of variance (ANOVA) followed by Tukey's HSD post hoc test. *$P < 0.05$, **$P < 0.01$, ***$P < 0.001$, and ****$P < 0.0001$. Source data are provided as a Source Data file.

China). FITC anti-mouse CD45 Antibody (#103107, 1:200), Pacific Blue™ anti-mouse/human CD11b Antibody (#101223, 1:100), APC/Cyanine7 anti-mouse F4/80 Antibody (#123117, 1:100), PE/Cyanine7 anti-mouse CD86 Antibody (#105013, 1:200), PE anti-mouse CD206 (MMR) Antibody (#141705, 1:100), PE-conjugated anti-mouse CD3 Antibody (#100205, 1:100), FITC-conjugated anti-mouse CD4 (#100406, 1:200), APC-conjugated anti-mouse CD8a Antibody (#100712, 1:200), Purified anti-HMGB1 Antibody (#651401, 1:100) were purchased from Biolegend. Mouse TNF-α, IL-6, IL-12p70 ELISA Kits and eFluor-450-conjugated anti-mouse Foxp3 antibody (#48-5773-82. 1:200) were obtained from Invitrogen. Alexa Fluor® 488 Anti-Calreticulin antibody [EPR3924] (ab196158, 1:100) was bought from Abcam.

## Apparatus

Transmission electron microscopy (TEM) was performed on a JEM-2100F microscope (JEOL Ltd., Japan). Powder X-ray diffraction (PXRD) measurements were carried out by using a Rigaku Ultima IV X-ray powder diffractometer with Cu K$\alpha$ radiation ($\lambda = 1.54056$ Å) over the range of $2\theta = 1.5 - 30.0°$ with a step size of 0.02° and 0.4 s per step. The BET surface area and pore size measurements were performed with N$_2$ adsorption/desorption isotherms at 77 K on a Micromeritics ASAP 2020 M instrument. Thermogravimetric analysis (TGA) was performed on a NETZSCH STA 409PC instrument by heating the polymer at 10 °C min$^{-1}$ to 800 °C under argon. The Fourier-transform infrared spectroscopy (FTIR) spectra (KBr) were recorded on a Bruker Tensor 27 Infrared spectroscope in transmission mode. The zeta potential and size were determined by Malvern Mastersizer 2000 (Zetasizer Nano ZS90, Malvern Instruments Ltd, UK). Fluorescence images were obtained with a Nikon A1R confocal laser scanning microscope. Inductively coupled plasma-atomic emission spectroscopy (ICP-AES) was carried out on Prodigy 7 (Leeman). PET images were taken with a micro-PET (Siemens Medical Solutions USA, Inc.). The $^1$H and $^{13}$C NMR spectra were recorded using a Bruker AV400 Spectrometer in solution at 400 MHz and 101 MHz, respectively. Proton ($^1$H) NMR information is given in the following format: multiplicity (s, singlet; d, doublet; t, triplet; q, quartet; qui, quintet; sept, septet; m, multiplet), coupling constant (s) (*J*) in Hertz (Hz), the number of protons. Carbon ($^{13}$C) NMR spectra are reported in ppm ($\delta$) relative to residual CDCl$_3$ ($\delta$ 77.00). X-ray photoelectron spectroscopy (XPS) analysis was performed on an Axis Ultra (Kratos Analytical Ltd.). Fluorescence-activated cell sorting (FACS) analysis was performed on BD FACSAria III and LSRFortessa flow cytometer.

## General synthetic route of Precursor Compound 1-2[38,39]

4-Iodobenzaldehyde (5.00 g, 21.55 mmol), *p*-toluene-sulfonic acid (0.82 g, 4.30 mmol), ethylene glycol (14.45 g, 215.50 mmol) and molecular sieves were suspended in CHCl$_3$ (50 mL), and the mixture was heated to 80 °C and stirred overnight. The reaction mixture was cooled to room temperature, washed with water and NaHCO$_3$ (1 M), and the aqueous layer was extracted with chloroform. The combined organic layer was dried over Na$_2$SO$_4$, filtered and evaporated in vacuo to remove the volatile compounds. The crude product was purified by column chromatography on silica gel using a mixture of hexane and dichloromethane (1:2, *v:v*) as eluant to afford compound 1 (0.75 g, 50%) as a white solid. $^1$H NMR (400 MHz, Chloroform-*d*) δ 7.72 (d, *J* = 8.3 Hz, 2H), 7.22 (d, *J* = 8.3 Hz, 2H), 5.76 (s, 1H), 4.10 (dd, *J* = 4.2, 2.2 Hz, 2H), 4.05 – 4.00 (m, 2H).

The *n*-butyllithium (*n*-BuLi) (4.3 mL, 6.90 mmol) in hexanes was added dropwise into a solution of 1,12-dicarbadodecaborane (200 mg, 1.38 mmol) in anhydrous THF (40 mL) in −78 °C under nitrogen. The reaction was stirred at −78 °C for 0.5 h and then warmed to room temperature for another 2 hours. CuCl (342 mg, 3.45 mmol) was added into the system quickly, and the reaction was stirred at room temperature for another 1 h. Compound 1 (952 mg, 3.45 mmol) in pyridine (3 mL) was added into the system quickly, and the reaction was stirred at 80 °C for another 12 h. The reaction mixture was poured into water and extracted with dichloromethane. The combined organic layer was dried over Na$_2$SO$_4$, filtered and evaporated in vacuo to remove the volatile compounds to afford crude compound **2**.

## General synthetic route of *p*-carborane-1,10-phenyl-dialdehyde

The trifluoroacetate (3 mL) was added into a solution of compound 2 in chloroform (20 mL) obtained in the previous step. The reaction was stirred at room temperature for 1 h and then the reaction mixture was poured into water and extracted with dichloromethane. The organic layer was washed with water and then dried over Na$_2$SO$_4$. After removal of the solvent, the crude product was purified by column chromatography on silica gel using a mixture of hexane and dichloromethane (1:2, *v:v*) as eluant to afford *p*-carborane-1,10-phenyl-dialdehyde (B-CHO) (0.89 g, 86%) as a white solid. $^1$H NMR (400 MHz, Chloroform-*d*) δ 9.97 (s, 2H), 7.73 (d, *J* = 8.5 Hz, 4H), 7.42 (d, *J* = 8.5 Hz, 4H), 2.67 (d, *J* = 195.0 Hz, 10H). $^{13}$C NMR (101 MHz, CDCl$_3$) δ 191.26, 141.94, 136.09, 129.40, 127.93, 29.68.

## Synthesis of the carborane-based covalent organic framework (B-COF)

Typically, 50 μL polystyrene microspheres (PS) suspension (2.5% w/w in water, 100 nm) was mixed with 150 μL acetic acid (12 M HAc) and sonicated for 30 min to form a homogeneous suspension for preparation. A mixture of 1,3,5-tris(4-aminophenyl) benzene (TAPB) (7 mg, 0.02 mmol), B-CHO (10.3 mg, 0.03 mmol), the acetic acid suspension (200 μL) and *o*-dichlorobenzene (2.5 mL) were reacted at room temperature for 8 hours. Next, 4-benz-carborane-aldehyde (2.2 μL, 0.02 mmol) was added to the reaction system to quench the reaction. After 1 hour, the products were isolated by centrifugation and washed with acetonitrile and methanol three times to generate PS-B-COF that appeared as faint yellow powders. Then, the newly prepared product in toluene (20 mL) was heated to reflux for 24 h under continuous stirring to dissolve the PS templates. B-COF was finally isolated by centrifugation and washed with acetonitrile and methanol until the supernatant liquid was colorless and dried at 50 °C for 24 h to generate B-COF.

## Synthesis of imiquimod loaded B-COF

To test the loading capacity, 100 mg of imiquimod and 40 mg of B-COF in ethanol were added to a glass vial. The mixture was sonicated for 1 hour at 50 °C and stirred at room temperature for another 24 hours. Imiquimod loaded B-COF was isolated by centrifugation and washed with dimethyl sulfoxide (DMSO) and ethanol. The supernatant of

imiquimod was detected by UPLC-MS to confirm the absence of imiquimod. Determination of the amount of imiquimod loaded on B-COF was assessed by UPLC-MS.

### Synthesis of boron capsule (the PEGylated imiquimod loaded B-COF)

The prepared imiquimod-loaded B-COF were added into the DSPE-PEG2000 solution and stirred at room temperature for 2 hours, residual DSPE-PEG2000 and DMSO were removed by dialysis (Molecular weight cut-off 10 kDa). Boron capsule was finally obtained after the freeze-drying process. PEGylated B-COF (PEG-B-COF), fluorescent boron capsule (Fluorescent PEGylated imiquimod loaded B-COF) and amino-boron capsule (Amino-PEGylated imiquimod loaded B-COF) were also synthesized according to a similar procedure as described above, except that DSPE-PEG was replaced by FITC-DSPE-PEG or DSPE-NH$_2$-PEG, respectively.

### Details of the thermal neutron source

All experiments were performed at the In-Hospital Neutron Irradiator (IHNI) based on a miniature neutron source reactor, a low power research reactor designed and manufactured by the China Institute of Atomic Energy (CIAE). The rated power of IHNI is 30 kW, the neutron flux is $1.9 \times 10^9$ cm$^{-2}$·s$^{-1}$ and the thermal neutron beam energy field of IHNI is <0.4 eV.

### Neutron facilitated imiquimod release from boron capsule

Boron capsule (B-COF = 1 mg/mL) in phosphate-buffered saline (PBS) was irradiated with a neutron beam at a neutron flux of $1.9 \times 10^9$/(cm$^2$·s) for 30 min. Then, the boron capsule was dialyzed (Molecular weight cut-off 10 kDa) at different times to remove the released imiquimod. The released imiquimod was analyzed by UPLC-MS. The cumulative drug release was calculated from the following equations.

$$\text{Cumulative release of drug} (\%) = \frac{V_e \sum_1^{n-1} C_i + V_0 C_n}{m_0} \times 100(\%) \circ \quad (1)$$

$V_e$: displacement volume of water; $V_0$: total volume of release medium; $C_i$: concentration of release solution at the $i$-th displacement sampling; $m_0$: the total mass of drug contained in boron capsule; n: number of water displacements.

### Quantification methods of imiquimod

The standard curve of imiquimod was prepared by dissolving pure imiquimod in water (1% DMSO) and the linear range was between 10 μM and 50 μM. The gradient elution of the UPLC-MS was set as: (1) 0-1 min, 99% H$_2$O and 1% MeOH; (2) 3-8 min, 65% H$_2$O and 35% MeOH; (3) 8-10 min 100% MeOH. The flow rate was 0.5 mL/min with an injection volume of 3 μL and the samples were diluted 100-fold for the UPLC-MS analysis.

### Monte Carlo calculation and molecular dynamics (MD) simulation

During the Monte Carlo simulation, Geant 4.10.7 was executed for the interaction of thermal neutrons with the B-COF microsphere. The single 1-μm B-COF sphere with the radius parameter set to 0.5 μm and the density set to 0.02 g/cm$^3$. A disc-shaped neutron beam with a radius of 0.5 μm was set to single energy (0.025 eV) and the total number of neutrons emitted from 1 cm to the center of the geometry was set to 100 million. A square-shaped neutron beam with a side length of 4 μm was set to single energy (0.025 eV) for 64 1-μm B-COF spheres. The total number of neutrons emitted from 1 cm to the center of the geometry was set to 1.6 billion. During molecular dynamics simulations, adaptive intermolecular reactive empirical bond-order (AIREBO) potential and the Lennard-Jones (LJ) potential were used for

the carbon-carbon interactions in graphene and carbon-lithium interactions, respectively[40].

### Cell culture assay

Murine melanoma B16F10 cells (1101MOU-PUMC000473) and colon carcinoma MC38 (1101MOU-PUMC000523) cells were obtained from National Infrastructure of Cell Line Resource (Beijing, China). Primary mouse bone-marrow-derived macrophages BMDMs was kindly provided from Prof. F. Shao's Lab (National Institute of Biological Sciences, Beijing)[41]. B16F10 and BMDMs were cultured in Dulbecco's modified eagle medium (DMEM) with 10% fetal bovine serum (FBS), 1% penicillin/streptomycin, and 1% glutamine at 37 °C under 5% CO$_2$ in a humidified incubator while colon carcinoma MC38 cell was cultured with Roswell Park Memorial Institute (RPMI)-1640 medium at the same condition.

### Cell viability assay

The cytotoxicity of the boron capsule and B-COF were evaluated in B16F10 and MC38 cells using the Cell Counting Kit-8 (CCK8). B16F10 and MC38 cells were treated with boron capsule and B-COF for 24 h, respectively. The untreated cell population was used as the reference to establishing 100 % cell viability.

### Cellular uptake study

B16F10 and MC38 cells with a density of $2 \times 10^4$ cells per well were seeded into 24 well culture plates. After 24 h interval, the culture medium containing fluorescent boron capsule (B-COF = 1 mg/mL) was added to each well. After 2 h and 24 h incubation respectively, the medium in each well was washed several times with PBS. After that, the cell nucleus was stained with DAPI (4,6-diamidino-2-phenylindole) and the fluorescence intensity was monitored by confocal fluorescence microscopy.

### Cellular boron concentration study

B16F10, MC38 and BMDM cells were incubated with boron capsule (B-COF = 1 mg/mL) for 2 and 24 h respectively. Next, the cells were rinsed with PBS three times to remove the boron contents in the culture medium. After that, trypsin solution was added to facilitate cell dissociation from their culture substrate. The lysates were collected and continued to digest crudely using a microwave accelerated reaction system (Mars; CEM), followed by dilution with deionized water. Cellular boron concentration was measured by ICP-AES according to the published method.

### Flow-cytometry analysis

B16F10 and MC38 cells were treated with boron capsule (B-COF = 1 mg/mL), B-COF (1 mg/mL), imiquimod (5 μg/mL) and culture medium for 24 h. Next, the cells were rinsed with PBS three times followed by neutron irradiation with a neutron flux of $1.9 \times 10^9$/(cm$^2$·s) for 10 min. After 24 h incubation, the cells were stained according to the Annexin V-FITC/PI Apoptosis Detection Kit and quantified by flow cytometry.

### Clone formation experiment

After neutron irradiation, different groups of B16F10 and MC38 cells were reseeded with appropriate dilutions to form colonies. Finally, colonies were fixed with glutaraldehyde (6.0% v/v), stained with crystal violet (0.5% w/v) and analyzed by the microscope.

### Synthesis of $^{89}$Zr-boron capsule

Amino-boron capsule (B-COF = 1 mg/mL) in Na$_2$CO$_3$/NaHCO$_3$ buffer solution (pH = 9.0) was prepared. p-isothiocyanatobenzyl-desferrioxamine (DFO-Bz-NCS) dissolved in dry DMSO (10 mM) was added to the amino-boron capsule stock solution. The above reaction was incubated for 1 h at 37 °C on an agitating heating block. The obtained

DFO conjugated amino-boron capsule solution was purified by ultrafiltration and centrifuged at 8000 rpm for 10 min five times. For the radiolabelling experiments, $^{89}Zr(C_2O_4)_2$ solution (100 μL, 2 mCi) was added into HEPES buffer (0.5 M, 400 μL) and the pH of the solution was adjusted to 7.0 by the addition of $Na_2CO_3$ (0.2 M). This mixture was added to the DFO conjugated amino-boron capsule solution stock solutions and reacted for 1 h at 37 °C on an agitating heating block[32]. The $^{89}Zr$-boron capsule was purified by centrifuged at 8000 rpm five times.

### PET imaging

B16F10 tumour-bearing mice were intratumourally injected with 50 μCi of $^{89}Zr$-boron capsule for dynamic positron emission tomography-computed tomography (PET-CT) imaging. Mice were anesthetized under a 2% isoflurane/oxygen atmosphere and imaged by a micro-PET/CT rodent model scanner.

### In vivo anticancer efficacy

6-8 week-old C57BL/6 female mice (Vital River Laboratories, China) were injected subcutaneously with $2 \times 10^6$ B16F10 cells or MC38 cells in the right flank. Length ($L$) and width ($W$) of the tumour were determined by digital calipers. The tumour volume ($V$) was calculated by the formula $V = 1/2 \times L \times W^2$. When the volume of the tumour reached a size of approximately $100 \pm 25$ mm$^3$, it is time for the PET imaging and therapeutic experiment. The bilateral syngeneic B16F10 and MC38 mouse models were established by injecting $2 \times 10^6$ and $1 \times 10^6$ cells into the right and left flank subcutaneous tissues of C57BL/6 mice on day 0 to mimic primary and distant tumours, respectively. When the primary tumours reached 75-125 mm$^3$ in volume, the mice were randomly distributed into 8 groups: boron capsule (B-COF = 2.5 mg/mL, 20 μL), PEG-B-COF (2.5 mg/mL, 20 μL), imiquimod (50 μg/mL, 20 μL) and PBS (20 μL) with (+) or without neutron irradiation. At 24 hours after the intratumoural injection, mice anesthetized with isoflurane (5%, 100 μL) were fixed onto the special-made module and irradiated with thermal neutron irradiation with a neutron flux of $1.9 \times 10^9/(cm^2 \cdot s)$ for 30 min. The tumour volume and body weight of each group was monitored as described in the Supplementary Information. When the tumour size reached 1200 mm$^3$ or the loss was > 20% of total body weight, the mice were removed from the experimental group and euthanized. In some cases, this limit has been exceeded the last day of measurement and the mice were immediately euthanized

### Single-cell RNA sequencing

For FACS of tumour-infiltrating lymphocytes, tumours were dissected on day 7 after BNCT. Cell clumps were removed through a 40-μm cell strainer to obtain single-cell suspensions. The suspension was centrifuged and the cell pellets were washed twice with PBS containing 1% BSA (FACS buffer). Lymphocytes were isolated by Percoll density-gradient centrifugation, washed and resuspended in the FACS buffer, stained with FITC-conjugated anti-mouse CD45 antibody (Biolegend). The Cell Apoptosis 7-AAD Detection Kit was used to determine cell viability during FACS analysis. CD45 + immune cells were then enriched by using a BD FACS Aria SORP flow cytometer. Cell viability was monitored in real-time during the preparation of the single CD45 + immune-cell suspension[32]. Single-cell libraries were generated via the Chromium Next GEM Single Cell 3' Reagent Kit v3.1. Ten thousand cells from each experimental group were barcoded and pooled using the 10x Genomics device. Samples were prepared following the manufacturer's protocol and sequenced on a Nova-PE150 NextSeq sequencer. The 10x Cell Ranger v.1.0. was used for barcode processing, UMI counting and aggregation of the sequencing runs. Genes detected in fewer than three cells across the dataset were also excluded, yielding a preliminary expression matrix of 10,069 cells. After obtaining the digital gene-expression data matrix, Seurat (version 3.2.0) was used for dimension reduction and clustering. The code used for the analysis of scRNA-seq data can be accessed from Abiosciences Co., Ltd. (https://www.abiosciences.com.cn)

### FACS analyses of tumour-infiltrating lymphocytes

Tumours were harvested, treated with 1 mg/mL collagenase I (Gibco, USA) for 1 h in a 37 °C water bath, and ground using the rubber end of a syringe. Cells were filtered through nylon mesh filters with the size of 40 μm and washed with PBS. Cells were further stained with the following fluorochrome-conjugated antibodies: CD45 (30-F11), CD11b (M1/70), F4/80 (BM8), CD86 (GL-1), CD206 (C068C2), CD3 (17A2), CD4 (GK1.5), CD8a (53-6.7) and Foxp3 (FJK-16s). BD LSRFortessa was used for cell acquisition and data analysis was carried out with FlowJo software (BD Biosciences, USA).

### Statistics and reproducibility

Student's $t$-test (two group comparison) and one-way ANOVA with Tukey's honest significant difference (HSD) post hoc test (multiple groups comparison) were performed to determine statistical significance between different groups. Data are presented as means ± SD. as indicated in the figure captions. Significance was denoted in figures as $*P < 0.05$, $**P < 0.01$, $***P < 0.001$, and $****P < 0.0001$.

### Reporting summary

Further information on research design is available in the Nature Portfolio Reporting Summary linked to this article.

## Data availability

Source data are provided with this paper. The raw sequence data reported in this paper have been deposited in the Genome Sequence Archive in National Genomics Data Center, China National Center for Bioinformation / Beijing Institute of Genomics, Chinese Academy of Sciences, under accession number CRA005110, that are publicly accessible at https://ngdc.cncb.ac.cn/gsa/browse/CRA005110.

The remaining data are available within the Article, Supplementary Information or Source Data file. Source data are provided with this paper.

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

## Acknowledgements

We thank Q. Sun, X. Liu and Y. Cui for single-cell RNA sequencing support; H. Lv and Y. Guo for assistance with fluorescence activated cell sorting (FACS); M. Yang for helpful discussions. The measurements of NMR, transmission electron microscope imaging and confocal imaging were performed at the Analytical Instrumentation Center of Peking University. The FACS was performed in the College of Life Sciences at Peking University. This study was funded by the National Nature Science Foundation of China (Grants No. 22225603, U1867209), the Ministry of Science and Technology of the People's Republic of China (Grant Nos. 2021YFA1601400), the Beijing Municipal Natural Science Foundation (Grant No. Z200018), and Changping Laboratory under the project number (2022C-07-01), the Central Guidance for Local Science and Technology Development Projects (No. 202138–03) to Z.L. and National Key R&D Program of China (2021YFA0909900) to Z. Gu.

## Author contributions

Z.L. and Z.Gu conceived the idea; Z.L. and Z.Gu supervised the project. Y.S., Z.Guo and Q.F. performed COF material synthesis, characterization and chemical analysis. X.S provided technical assistances and valuable suggestions. Zh.Z. carried out the Monte Carlo calculations. W.S. and J.S. carried out COF material calculation. Y.S. and Z.Guo performed all other experiments. Zi.Z. and T.L. provided BNCT technical assistances and performed SERA simulation. Y.S., Z.Guo, Q.F., J.W., Z.Gu and Z.L. prepared the manuscript. All authors discussed the results and commented on the manuscript.

## Competing interests

Z.L. and Y.S. have applied for patents (ZL20211342701.X, the synthesis and applications for a covalent organic framework) related to this study. Z. Gu is scientific cofounder of ZenCapsule Inc. and ZCapsule Inc. The remaining authors declare no other competing interests.
