## [Peer Review File · Nature Communications]

Localized Nuclear Reaction Breaks Boron Drug Capsules and Augments Cancer ImmunotherapyEditorial Note: Reviewer names have been redacted from their review at their request.

REVIEWER COMMENTS

Reviewer #1 (Remarks to the Author): with expertise in nanomedicine, cancer immunology

The manuscript by Shi et al results very interesting and innovative, with clear and positive results. Indeed, the latest advances in the field, such as proton-therapy or boron neutron capture therapy (BNCT), have demonstrated a very significant improvement with respect to traditional radiotherapy providing new solutions for patients with solid tumors. In this context the development of nano-/microparticles to enhance the efficacy of the therapy and reducing off-target effects results of foremost importance.

The current work presents a first part dedicated to the development and characterization of the “boron drug capsules” with comprehensive physicochemical studies and details. This is a high-quality work with a clear presentation and nice graphical presentation. The researchers use state-of-the-art techniques for this work, which encourage the publication of the work in a high-impact journal. Instead, the biological study is limited. Again, the authors use state-of-the-art techniques, but the study is a bit narrow with respect to the cell death and immunomodulatory effect of the nanoparticles. In particular, I miss further detail of innate immunity in the tumor microenvironment and study of systemic immunity to explain the abscopal effect. Also study of pharmacokinetics (systemic release of the payload) or possible immunotoxicity related effects must be discarded (as high levels of pro-inflammatory cytokines have been found in systemic circulation). These are major concerns which must be addressed for the publication of this manuscript in a high-impact journal.

Furthermore I recommend to address the following minor concerns:

1 - From the beginning it is not really clear which are the differences between the particles. I understand that “B-COF” are the “blank” particles, made of the framework and boron ions, and the “boron capsules” are the B-COF loaded with imiquimod and coated with DSPE-PEG. I think this should be stated more clearly.

2 – According with the previous comment, if this is right, in the in vivo experiments the right comparison between the capsules with or without imiquimod should have been the empty B-COF coated with DSPE-PEG versus B-COF loaded with imiquimod and coated with DSPE-PEG. As it stands the difference observed between the 2 groups in terms of efficacy could be due to a better stability and dispersibility, given that the empty particles (B-COF) seem to have immunological effects.

3 – With regard to the biological effect, the authors mention mechanisms of pyroptosis or immunogenic cell death which have not been properly investigated in their work. No experimental evidence is provided. I recommend the evaluation of markers of pyroptosis or ICD. And also discard other mechanisms such as apoptosis or senescence, as these mechanisms have been related with radiotherapy or immunotherapy treatments, in the context of abscopal effect, resistance or systemic antitumoral immunity. Thus, a detailed study in this regard for these particular boron capsules would improve the relevance of the present work.

- With regard to the study of innate immunity, first I miss a clear quantification of particle accumulation in immune cells, in particular in macrophages and dendritic cells, within the tumor microenvironment. Particles probably accumulate in these phagocytic cells. While BCNT would kill these cells, imiquimod should activate their pro-inflammatory activity. Indeed, a decrease in myeloid cells is observed in treated these tumors (fig. 5d). Thus, these are relevant aspects to be explored in detail.

- Regarding the study of the tumor microenvironment, I recommend moving from the supp. info. to the main manuscript the results related to myeloid cells. Innate immunity is clearly activated by imiquimod and BCNT and percentage of myeloid cells in treated tumors is still high. Thus, more explanation

about role of macrophages and neutrophils is needed. I miss an explanation about the difference between M2 macrophages and TAMs in the analysis. Also about the gating strategy for M1 macrophages, as I consider that all CD86 high cells should be considered. I can imagine that M1/M2 ratio is calculated by CD86+/CD206+, but this is not clear and should be clearly explained.

- Regarding the levels of circulating cytokines after treatment the authors claim that "This result indicates that a systematic anti-tumor immune response emerges, as IL-12 plays important role in activating natural killer cells, and IL-6 and TNF- α are critical". Could it be also a reflection of some systemic toxicity? more studies are needed in this regard.

- In fig. 5, the immunological remodeling of the tumor microenvironment is compared for PBS+N versus B-COF+N. So, I understand that RNA-seq analysis was only performed with the group not containing imiquimod. This is a rather strange choice, as it would make sense to compare RNA changes also the therapy with or without adjuvant (in addition to the FACS analysis).

- There's a discrepancy between fig. 4a's legend and the methods: particles injected intravenously versus intratumorally.

Reviewer #2 (Remarks to the Author): with expertise in boron neutron capture therapy

This manuscript reported a boron capsule based on carborane-based covalent organic framework to achieve antitumor effect through BNCT and controlled drug release triggered by localized nuclear reaction. The author brought up an appealing topic on BNCT-induced antitumor immune activation. Overall, based on the current data and analysis, this manuscript cannot reveal the mechanism of BNCT-induced antitumor immunity clearly.

1. As shown in Figure 4, it is shown that the boron capsule has favorable tumor targeting after intravenous administration. However, it is known that B16F10 tumor-bearing mice grows rapidly, and the PEG-modified micro/nano-particles could be transported into tumor tissue via EPR effect, while EPR is not obvious in clinical patients. In addition, intra-tumoral injection in this study increase the intra-tumoral retention of the boron capsule, but considering the larger particle size, would it adversely affect its diffusion, penetration and cellular uptake in tumor tissue? In contrast, the small molecule BPA used in BNCT therapy have certain tumor targeting and cellular uptake. It is recommended that the experimental control group using BSH or BPA be investigated in the experiment.

2. The authors emphasize the value of appropriate timing of sufficient adjuvants in tumor therapy in Introduction and in the Drug Release section, however this superiority of boron capsules has not been clearly demonstrated by the experiment results. When is the perfect timing for drug release? And how to determine this timing? Does the appropriate timing mean simultaneously administration of BNCT and imiquimod? What differences would it make when BNCT and immune adjuvants were administrated successively or separately since the boron capsules were given intratumorally? More importantly, the authors need to add the control group as B-COF+imiquimod+N for the stronger illustration on this issue,

3. What pathways do BNCT utilize to remodel tumor immune microenvironment including T cells activation and inhibition on myeloid cells recruitment? As the author indicated in the section of Pyroptotic-like cell death with neutron irradiation, was it due to the pro-inflammatory effect of BNCT-induced pyroptotic cell death? It would be better to explore a bit deeper to conduct relevant experiments on T cells in vitro.

4. Please explain or at least discuss about the reason why intratumoral administration of imiquimod alone exerted almost no inhibitory effect on tumor growth, since imiquimod administration seems to greatly increase M1/M2 ratio in tumors and serum cytokines including TNF- α , IL-12 and IL-6.

5. The authors gave a garbled account of the heterogeneous tumor killing effect brought by BNCT. Was the heterogeneity caused by the heterogeneous distribution of boron capsule in tumors? Does the author believe that tumor cells apoptosis which was indicated by the detection of DNA fragments via TUNEL staining can only be contributed by BNCT rather than the antitumoral cytotoxic T cells? Besides, in Figure 4i and 4j, it is suggested to scan the entire slices to give a full impression on nuclei shrinkage and DNA degradation in tumor tissues as the authors demonstrated in this section,

6. Please reconstruct the manuscript more carefully to avoid misleading or unnecessary mistakes. For example, the administration pathway of boron capsule stated in the figure legend of Figure 4a is inconsistent with the statement in the main text. Significance labeling are missing in Figure 6. It's confused that whether the single-cell RNA sequencing was analyzed on mice administrated with immune adjuvants or not. The schematic illustration indicates Boron capsule injection while the treatment was described as B-COF+neutron in the main text.

Reviewer #3 (Remarks to the Author): with expertise in boron neutron capture therapy

This manuscript reports original and contributory findings of great relevance to the advancement of BNCT. The science is sound. The manuscript addresses the very important issue of attaining a systemic/abscopal effect with a local therapy. In my opinion, combined therapies are the best strategy for cancer therapy and in that sense the approach reported herein combines BNCT with immunotherapy, a very promising strategy. The manuscript reports a large amount of very useful data that support the conclusions. The development of a neutron activated boron capsule that synergizes BNCT and the controlled release of immune adjuvants to favor an antitumor response, is of great significance in the field.

However, before I can recommend publication, the authors should address the following issues:

- 1) Discuss the proposed strategy (benefits and limitations) considering the target pathology and the organs at risk.
- 2) Discuss tumor selectivity in terms of boron carrier uptake.
- 2) Describe/discuss the radiotoxic effects in normal tissues of the treatment described in the manuscript.
- 3) Describe the in vivo irradiations, features of the neutron source and neutron spectrum, animal shielding, irradiation dose components, absorbed dose delivered to tumor and to dose-limiting tissues... These data will allow the reader to interpret the data within the context of the vast existing literature on BNCT.

[redacted]

Reviewer #4 (Remarks to the Author): with expertise in cancer immunology/immunotherapy

GENERAL COMMENTS:

The manuscript presents the investigation "...of whether a carborane-based covalent organic framework (B-COF) can be based to develop a boron "capsule" of immune adjuvants for concurrent BNCT and immunotherapy". To test their hypothesis the authors performed in vitro and in vivo experiments that are adequate but some critical issues must be addressed. The issues are listed below.

MAJOR CONCERNS:

- To determine immune cell subtypes based on differentially expressed markers, which statistical test was used in Seurat for the differential expression analysis?

- Other concerns regarding the scRNA-seq portion of the study are the availability of the data (raw and Seurat processed) and the availability of the code used. Those are essential data and material to ensure that others can access and reproduce the results.
- The authors use bilateral tumor implantation in mice as a model of metastasis by treating one of the tumors while keeping the other side treatment intact to mimic the lack of irradiation in metastatic lesions in patients. However, in my opinion, this is not the most suitable murine metastatic cancer model. There are multiple protocols to induce metastasis in mice (e.g.: intravenous injection for lung colonization, one of the sites of colorectal cancer metastasis) and even models that develop spontaneous metastasis. Why did the bilateral model choose as a model of metastasis other than other models that better mimic metastatic disease and systemic irradiation effects? This is a critical experiment and the most adequate model impacts biological interpretation and relevance of the findings.
- The comparisons between the bilateral tumors only show the cytokines and cell types proportions for the "primary tumors". It is critical for biological interpretation that authors include for the "distant tumors" the results for serum cytokines and immune subtypes as shown for the "primary tumors" in Figures 6d-g. In the same way, the graphs demonstrating tumor growth curves for the "primary tumors" are essential.

MINOR CONCERNS:

- Figure 1i refers to more than one panel. I recommend the authors label all panels, thus Figure 1i will be broken into i, j and k. Also, the scale for the bars is only shown in one panel and I recommend adding that to all.
- In Figure 4b, the scale bars are missing the size. The same happens to the other panels in Figure 4, where the size is only shown in the last image of each panel. Just to facilitate readers' comprehension, I recommend adding the size to all images.
- Another issue with Figure 4 is the order of the panels. I recommend the authors place the panels in alphabetical order.
- In Figure 5d, the proportions of myeloid, B and NK cells are shown grouped. I recommend breaking them to show each individual proportion instead of grouping the subtypes to facilitate results interpretation.

RESPONSES TO THE COMMENTS

Please kindly find our point-by-point response to reviewers' comments below.

Reviewer #1

The manuscript by Shi et al results very interesting and innovative, with clear and positive results. Indeed, the latest advances in the field, such as proton-therapy or boron neutron capture therapy (BNCT), have demonstrated a very significant improvement with respect to traditional radiotherapy providing new solutions for patients with solid tumours. In this context the development of nano-/microparticles to enhance the efficacy of the therapy and reducing off-target effects results of foremost importance. The current work presents a first part dedicated to the development and characterization of the “boron drug capsules” with comprehensive physicochemical studies and details. This is a high-quality work with a clear presentation and nice graphical presentation. The researchers use state-of-the-art techniques for this work, which encourage the publication of the work in a high-impact journal. Instead, the biological study is limited. Again, the authors use state-of-the-art techniques, but the study is a bit narrow with respect to the cell death and immunomodulatory effect of the nanoparticles. In particular, I miss further detail of innate immunity in the tumour microenvironment and study of systemic immunity to explain the abscopal effect. Also study of pharmacokinetics (systemic release of the payload) or possible immunotoxicity related effects must be discarded (as high levels of pro-inflammatory cytokines have been found in systemic circulation). These are major concerns which must be addressed for the publication of this manuscript in a high-impact journal.

- We thank the reviewer for his/her encouraging comments that our work is very interesting and innovative. The manuscript has been revised accordingly.

1. From the beginning it is not really clear which are the differences between the particles. I understand that “B-COF” are the “blank” particles, made of the framework and boron ions, and the “boron capsules” are the B-COF loaded with imiquimod and coated with DSPE-PEG. I think this should be stated more clearly.

- We thank the reviewer for the suggestion and apologize for the unclear presentation in the original submission. B-COF refers to the “blank” particle, composed of the carborane-based covalent organic framework. Boron capsule refers to the PEGylated imiquimod loaded B-COF. In the original submission, we also used “B-COF” to describe PEGylated empty B-COF, which has been corrected by “PEG-B-COF” in the revised manuscripts.

2. According with the previous comment, if this is right, in the in vivo experiments the right comparison between the capsules with or without imiquimod should have been the empty B-COF coated with DSPE-PEG versus B-COF loaded with imiquimod and coated with DSPE-PEG. As it stands the difference observed between the 2 groups in terms of efficacy could be due to a better stability and dispersibility, given that the empty particles (B-COF) seem to have immunological effects.

- We apologize for the unclear presentation. In the original submission, we inappropriately use “B-COF” to describe the PEGylated empty B-COF in the cellular experiments and animal experiments, which has been corrected as “PEG-B-COF” accordingly. In fact, PEG-B-COF has as good dispersibility and stability as PEG-B-COF loaded with imiquimod (boron capsule).

3. With regard to the biological effect, the authors mention mechanisms of pyroptosis or immunogenic cell death which have not been properly investigated in their work. No experimental evidence is provided. I recommend the evaluation of markers of pyroptosis or ICD. And also discard other mechanisms such as apoptosis or senescence, as these mechanisms have been related with radiotherapy or immunotherapy treatments, in the context of abscopal effect, resistance or systemic antitumoural immunity. Thus, a detailed study in this regard for these particular boron capsules would improve the relevance of the present work.

- We thank the reviewer for the constructive advice. As suggested, considering that immunogenic cell death (ICD) is commonly characterized by the release of damage-associated molecular patterns (DAMPs), including calreticulin (CRT) and high-mobility group box 1 protein (HMGB1), we evaluated DAMPs to investigate the immunogenic cell death of BNCT.

As shown in **Figure R1**, BNCT induced a significantly higher degree of CRT exposure and HMGB1 release while the other groups were minimal. In addition, the PEG-B-COF+N group showed a comparable degree of CRT exposure and HMGB1 release to the boron capsule+N group. The above findings indicated that ICD was successfully induced by the BNCT of B-COF through thermal neutron irradiation.

Figure R1. Representative immunofluorescence staining of calreticulin (CRT) exposure (A) and high-mobility group box 1 protein (HMGB1) release (B) are shown. Tumour were harvested 7 days after the indicated treatment.

We have also performed pyroptosis experiments on B16F10 and observed a significant pyroptotic morphology, however, the pore-forming activity of the gasdermin D-N domain (GSDMD-N) released

from B16F10 cells treated with BNCT was difficult to detect, which was consistent with the low expression of GSDMD in the murine melanoma cell line B16F10. We have been trying to detect the markers of pyroptosis in the last 8 months, but found that it is rather complicated, therefore recommend using “pyroptosis-like” in this work for better accuracy. We completely agree with the reviewer that understanding the mechanisms of BNCT-induced inflammation is of great importance. We will continue working on this problem, and may solve this challenge in a few years.

4. With regard to the study of innate immunity, first I miss a clear quantification of particle accumulation in immune cells, in particular in macrophages and dendritic cells, within the tumour microenvironment. Particles probably accumulate in these phagocytic cells. While BCNT would kill these cells, imiquimod should activate their pro-inflammatory activity. Indeed, a decrease in myeloid cells is observed in treated these tumours (fig. 5d). Thus, these are relevant aspects to be explored in detail.

- We thank the reviewer for raising this very interesting question. As suggested, we have investigated the cellular uptake of boron capsules and cell death after neutron irradiation in primary mouse bone-marrow-derived macrophages (BMDMs). As shown in **Figure R2A**, BMDMs incubated with FITC fluorophore-conjugated boron capsule (1 mg/mL) for 24 hours showed that boron capsules can accumulate in phagocytic cells. We then performed a statistical analysis by confocal microscopy, and the average number of boron capsules per cell after 24 h of incubation was 9.2 ± 5.0 for BMDMs compared with 45.7 ± 9.9 for B16F10 tumour cells (**Figure R2B**), indicating that the cellular uptake of BMDMs was significantly lower than that in tumour cells.

Figure R2. (A), Representative confocal images of IBMM cells are shown after being treated by FITC fluorophore-conjugated boron capsule (1 mg/mL) for 24 hours. (B), Number of boron capsule per cell in the giving field of view (FOV) by confocal microscopy. Data are shown as mean \pm s.d. (n = 10). (C), The cell death rate was calculated as Annexin V–FITC+/PI+ cells and Annexin V–FITC+/PI– cells.

In scRNA-Seq analysis, we did observe the reduction in the proportion of myeloid cells, but also observed an increase in the proportion of CD4+ T cells, CD8+ T cells and NK cells, suggesting that BNCT can adjust the immune-cell population infiltrated into the tumour tissue and remodel the tumour microenvironment. Nevertheless, as the portion of tumour cells are often notably more than immune cells in tumors, we would suggest that boron capsules are more likely to be endocytosed by tumour cells rather than immune cells.

5. Regarding the study of the tumour microenvironment, I recommend moving from the supp. info. to the main manuscript the results related to myeloid cells. Innate immunity is clearly activated by imiquimod and BCNT and percentage of myeloid cells in treated tumours is still high. Thus, more explanation about role of macrophages and neutrophils is needed. I miss an explanation about the difference between M2 macrophages and TAMs in the analysis. Also about the gating strategy for M1 macrophages, as I consider that all CD86 high cells should be considered. I can imagine that M1/M2 ratio is calculated by CD86+/CD206+, but this is not clear and should be clearly explained.

- We agree with the reviewer that the study of the tumour microenvironment is important, yet the data may be oversized if put in the main manuscript. We also thank the reviewer for pointing out the missing experiment details. As suggested, the gating strategy for M1/M2 macrophages, percentages of M1 and M2, as well as the ratio of the M1 to M2 have been added accordingly (**Figure R3**).

As the major cellular component of innate immune responses, myeloid cells such as monocytes or granulocytes (especially neutrophils) can be recruited and promote inflammation, in case of tissue damage. scRNA-Seq analysis showed that BNCT increased the percentage of neutrophils from 12.45% to 32.92% compared to PBS+neutron-treated tumours. These results highlight the immunostimulatory potential of BNCT.

The difference between TAMs and M1/M2 macrophages is that TAMs can be polarized into different phenotypes, and a variety of stimuli may shift their polarization to different inflammatory profiles according to local conditions. M1 and M2 macrophages coexisted as key components of the tumour microenvironment, M1-macrophages are defined as pro-inflammatory cells involved in killing cancer cells, while M2-macrophages are known to promote metastasis and immunosuppression.

Figure R3. (A), Representative gating strategies of M1 macrophage (CD206-CD11b+F4/80+CD45+) and M2 macrophage (CD206+CD11b+F4/80+CD45+). (B-G), Quantification of infiltrating CD3+ (B), CD4+ (C), CD8+ (D), Foxp3+ (E), M1 macrophage (F) and M2 macrophage (G) immune cell and M1/M2 ratio (H) from the distant B16F10 tumours (n = 4).

6. Regarding the levels of circulating cytokines after treatment the authors claim that “This result indicates that a systematic anti-tumour immune response emerges, as IL-12 plays important role in activating natural killer cells, and IL-6 and TNF- α are critical”. Could it be also a reflection of some systemic toxicity? more studies are needed in this regard.

- As suggested, a systematic evaluation has been performed to evaluate the safety issues of boron capsules after BNCT, including the routine blood test and Hematoxylin and Eosin (H&E) staining assay of hearts, livers, spleens, lungs and kidneys. Tumour-bearing mice treated with PBS, imiquimod, PEG-B-COF, and boron capsules, respectively, were sacrificed on day 14.

As shown in **Figure R4**, the routine blood test of the boron capsule+N and PEG-B-COF+N group showed negligible cytotoxicity. In addition, all liver and kidney function parameters were within the normal range, reflected by the results of blood biochemical tests. In addition, as shown in **Figure R5**, no abnormalities were observed in the major organs of all experimental groups. Based on these experimental results, the boron capsule shows biocompatibility and safety for the BNCT.

Figure R4. The blood routine and biochemical test of B16F10 tumour-bearing mice treated with Neurtron (N) alone, Imiquimod, PEG-B-COF + N and Boron Capsule + N at day 14 (n = 4). Data are means \pm s.d..

Figure R5. Representative H&E staining of major organs with different treatments (N = 3).

7. In fig. 5, the immunological remodeling of the tumour microenvironment is compared for PBS+N versus B-COF+N. So, I understand that RNA-seq analysis was only performed with the group not

containing imiquimod. This is a rather strange choice, as it would make sense to compare RNA changes also the therapy with or without adjuvant (in addition to the FACS analysis).

- We apologize for the unclear presentation in the original submission. The CD45+ immune cells from B16F10 tumours treated by PBS+neutron and PEG-B-COF+neutron (PEGylated empty B-COF) were subjected to single-cell RNA sequencing (scRNA-Seq) analyses. The purpose of the RNA-seq analysis experiment was to obtain a full picture of BNCT-triggered immunological remodeling in the tumour microenvironment. The experimental results showed that BNCT remodeled tumour immune microenvironment, revealing the anti-tumour immunological effect of BNCT.

8. There's a discrepancy between fig. 4a's legend and the methods: particles injected intravenously versus intratumourally.

- We apologize for the mistake and have made the correction accordingly in the revised manuscript.

Reviewer #2

This manuscript reported a boron capsule based on carborane-based covalent organic framework to achieve antitumour effect through BNCT and controlled drug release triggered by localized nuclear reaction. The author brought up an appealing topic on BNCT-induced antitumour immune activation. Overall, based on the current data and analysis, this manuscript cannot reveal the mechanism of BNCT-induced antitumour immunity clearly.

- We thank the reviewer for his/her comments that our work is appealing and apologize for the unclear presentation about BNCT-induced antitumour immunity. The manuscript have been revised accordingly.

1. As shown in Figure 4, it is shown that the boron capsule has favorable tumour targeting after intravenous administration. However, it is known that B16F10 tumour-bearing mice grows rapidly, and the PEG-modified micro/nano-particles could be transported into tumour tissue via EPR effect, while EPR is not obvious in clinical patients. In addition, intra-tumoural injection in this study increase the intra-tumoural retention of the boron capsule, but considering the larger particle size, would it adversely affect its diffusion, penetration and cellular uptake in tumour tissue? In contrast, the small molecule BPA used in BNCT therapy have certain tumour targeting and cellular uptake. It is recommended that the experimental control group using BSH or BPA be investigated in the experiment.

- We agree with the reviewer that the EPR effect may be inconsistent in patients. Therefore, boron capsules were administered by intratumoural injection, not intravenous injection. We apologize for the mistake in the original submission. As shown in **Figure 4 (Figure R6A)**, dynamic positron emission tomography-computed tomography (PET-CT) 3D projection images showed that [⁸⁹Zr]boron capsules have good retention and relatively good dispersion in tumours.

In addition, we have also studied the cellular uptake of boron capsules *in vitro* and in tumors. For the cellular uptake of boron capsules, as shown in **Fig. 3B (Figure R6B)**, rapid cell uptake was observed in the cytoplasm within 2 h and significantly more boron capsules were found in both cell lines at 24 h post-incubation. For the tumour tissue uptake of boron capsules, the TEM characterization shows that boron capsules could be taken up into tumour cells and distributed in the cell cytosol in B16F10 tumour tissue (**Figure R6C**).

Figure R6. (A), Dynamic positron emission tomography-computed tomography (PET-CT) 3D projection images of B16F10 tumour-bearing mice at the indicated time points after intravenous injection of [^{89}Zr] boron capsule. (B), Representative confocal images of B16F10 cells are shown after being treated by FITC fluorophore-conjugated boron capsule (1 mg/mL) for 2 hours and 24 hours, respectively. (C), Representative TEM images show the excessive cyto-distribution of boron capsule in B16F10 tumour tissue.

As suggested, a comparison BNCT study using BPA has been performed to investigate the limitation of BNCT and the potential advantages of boron capsule-induced combinational BNCT and immunotherapy. C57BL/6 mice engrafted subcutaneously with the B16F10 cancer cells were intravenously injected with BPA (200 mg/kg, for bringing the boron concentration of tumour consistent with boron capsule when irradiated) on day 6 followed by sequential neutron irradiation on day 7. As shown in **Figure R7**, mice treated with BPA+neutron irradiation behaved similarly to the boron capsule+neutron irradiation-treated mice. In contrast, compared with the boron capsule+neutron irradiation-treated group, mice treated with BPA+neutron irradiation behaved similarly to the PEG-B-COF+neutron-treated mice and showed normal and aggressive tumour growth. These suggest that as the “effective killing radius” of BNCT is highly localized (<10 μm , about the diameter of a cell), the neutron-induced release of an immune adjuvant may be essential to achieve comprehensive tumour treatment. Besides, this bystander effect can be promoted by immune adjuvants, as BNCT with boron capsules shows significantly better treatment efficacy than that of BNCT with PEG-B-COF and BPA.

Figure R7. Treatment assays of primary tumour and distant tumour growth in mice treated by BPA (A), PEG-B-COF and boron capsule (B) with neutron irradiation ($n = 6$). Data shown as mean \pm s.d. Data in Figure R7B were adapted from Figure 4E & 6H in the manuscript.

2. The authors emphasize the value of appropriate timing of sufficient adjuvants in tumour therapy in Introduction and in the Drug Release section, however this superiority of boron capsules has not been

clearly demonstrated by the experiment results. When is the perfect timing for drug release? And how to determine this timing? Does the appropriate timing mean simultaneously administration of BNCT and imiquimod? What differences would it make when BNCT and immune adjuvants were administered successively or separately since the boron capsules were given intratumorally? More importantly, the authors need to add the control group as B-COF+imiquimod+N for the stronger illustration on this issue.

- As suggested, we have performed the experiments that giving BNCT and immune adjuvants sequentially (PEG-B-COF-imiquimod+N) or separately (PEG-B-COF+imiquimod+N, imiquimod+PEG-B-COF+N) to compare the treatment efficacy. As High-energy daughter nuclei generated from BNCT can effectively break not only the DNA of tumor cells but also the boron capsule, the payload would gradually release to keep stimulating the immunity of tumour microenvironment in a relatively long term.

As shown in **Figure R8**, there was no significant difference in primary tumour therapeutic efficacy when BNCT and immune adjuvants were administered sequentially or separately. Of note, the group treated by boron capsule+neutron irradiation showed the best distant tumour suppression. The group treated by PEG-B-COF-imiquimod+N showed similar therapeutic efficacy to the group treated by the PEG-B-COF+imiquimod+N or imiquimod+PEG-B-COF+N, and showed a moderate distant tumour inhibition. These suggest that the neutron-induced release of an immune adjuvant may be of importance for tumour treatment. We agree with the reviewer that timing may not be that important, and have made the correction in the revised manuscript.

Figure R8. Therapeutic assays of primary tumour (A) and distant tumour (B) growth in mice treated by PBS, PEG-B-COF followed with imiquimod, PEG-B-COF with imiquimod simultaneously, imiquimod followed with PEG-B-COF and boron capsule with neutron (N) irradiation ($n = 6$). Data shown as mean \pm s.d. (two-tailed unpaired Student's t-test, **** $P < 0.001$).

3. What pathways do BNCT utilize to remodel tumour immune microenvironment including T cells activation and inhibition on myeloid cells recruitment? As the author indicated in the section of Pyroptotic-like cell death with neutron irradiation, was it due to the pro-inflammatory effect of BNCT-induced pyroptotic cell death? It would be better to explore a bit deeper to conduct relevant experiments on T cells *in vitro*.

- We agree with you that the elucidation of the immune pathways behind BNCT is a long-standing goal of researchers, and our lab is systematically working on this, but it is a challenging and time-consuming

task. The scRNA-Seq analysis suggested that BNCT remodels tumour immune microenvironment primarily by adjusting the components of immune cells, enhancing tumour T-cell infiltration and the expression of related pro-inflammatory genes. Based on the current experimental results, we concluded that the pro-inflammatory effect of BNCT includes the transformation of cell membrane permeability, calreticulin (CRT) exposure, and high-mobility group box 1 protein (HMGB1) release (Figure P1). The antitumour immune function of pyroptosis has been revealed in cancer treatment according to our published literature¹, suggesting that pyroptosis-induced inflammation triggers robust antitumour immunity and can synergize with checkpoint blockade. We have been trying to detect the markers of pyroptosis in the last 8 months, but found that it is rather complicated, therefore recommend using “pyroptosis-like” in this work for better accuracy. We completely agree with the reviewer that understanding the mechanisms of BNCT-induced inflammation is of great importance. We will continue working on this problem and may solve this challenge in a few years.

4. Please explain or at least discuss about the reason why intratumoural administration of imiquimod alone exerted almost no inhibitory effect on tumour growth, since imiquimod administration seems to greatly increase M1/M2 ratio in tumours and serum cytokines including TNF- α , IL-12 and IL-6.

- We agree with the reviewer that the administration of imiquimod has therapeutic efficacy, as we observed tumour suppression in the mice treated with imiquimod with neutron irradiation in the first 7 days. In comparison, boron capsule+neutron irradiation-treated mice had a pronounced growth delay for over 40 days, which is much longer than that of imiquimod treatment for the aggressive B16F10 tumours. As shown in **Figure R9**, similar inhibition on distant tumour growth was observed with boron capsule+neutron irradiation-treated mice and PEG-B-COF+neutron irradiation-treated mice. On the contrary, mice treated with imiquimod+neutron irradiation and PBS+neutron irradiation showed negligible tumour growth inhibition. The absence of BNCT makes it difficult for imiquimod to provide sufficient immune activation during thermal neutron irradiation, thus the tumour suppression efficiency was restricted at a late time point.

Figure R9. (A–D), Treatment assays of tumour growth in mice treated with boron capsule+neutron for 7 days post irradiation. The mice were treated by boron capsule, B-COF, imiquimod or PBS with (+N) or without neutron irradiation ($n = 6$). Data shown as mean \pm s.d.

5. The authors gave a garbled account of the heterogeneous tumour killing effect brought by BNCT. Was the heterogeneity caused by the heterogeneous distribution of boron capsule in tumours? Does the author believe that tumour cells apoptosis which was indicated by the detection of DNA fragments via TUNEL staining can only be contributed by BNCT rather than the antitumoural cytotoxic T cells? Besides, in Figure 4i and 4j, it is suggested to scan the entire slices to give a full impression on nuclei shrinkage and DNA degradation in tumour tissues as the authors demonstrated in this section.

- We thank the reviewer for the inspiring advice and apologize for the unclear presentation in the original submission. We agree with the reviewer that the heterogeneous tumour killing effect brought by BNCT attributes to both BNCT and the activation of the antitumoural cytotoxic T cells. The physical damage occurs in BNCT at the moment of thermal neutron irradiation, while it takes time to induce protective immune responses including recruitment of immune cells, secretion of cytokines, and provocation of a potent anti-tumour immune response. DNA fragments were detected 24 hours after BNCT and showed remarkable heterogeneity within the tumour tissue, consistent with the distribution of boron capsules. We would suggest that the majority of DNA heterogeneity fragmentation was caused by BNCT at early time points, and with the activation of the immune system, cytotoxic T lymphocytes also begin to participate in tumour suppression. In addition, as suggested, the entire slices with the full impression of nuclei shrinkage and DNA degradation in tumour tissues of all groups have been shown in **Figure R10**.

Figure R10. (A), Hematoxylin-Eosin (HE) analysis of entire slices of B16F10 tumour at 7 days after the indicated treatment. (B), DNA degradation assay in tumours at 1 day after the indicated treatment. Representative confocal images of tumour slices are shown. DNA fragments were assayed with terminal deoxynucleotidyl transferase dUTP nick end labeling (TUNEL, green fluorescence) and the nuclei were stained with DAPI (blue).

6. Please reconstruct the manuscript more carefully to avoid misleading or unnecessary mistakes. For example, the administration pathway of boron capsule stated in the figure legend of Figure 4a is inconsistent with the statement in the main text. Significance labeling are missing in Figure 6. It's confused

that whether the single-cell RNA sequencing was analyzed on mice administrated with immune adjuvants or not. The schematic illustration indicates Boron capsule injection while the treatment was described as B-COF+neutron in the main text.

- We thank the reviewer for the constructive suggestion and the corrections have been made accordingly.

Reviewer #3 (Remarks to the Author): with expertise in boron neutron capture therapy

This manuscript reports original and contributory findings of great relevance to the advancement of BNCT. The science is sound. The manuscript addresses the very important issue of attaining a systemic/abscopal effect with a local therapy. In my opinion, combined therapies are the best strategy for cancer therapy and in that sense the approach reported herein combines BNCT with immunotherapy, a very promising strategy. The manuscript reports a large amount of very useful data that support the conclusions. The development of a neutron activated boron capsule that synergizes BNCT and the controlled release of immune adjuvants to favor an antitumour response, is of great significance in the field. However, before I can recommend publication, the authors should address the following issues.

- We thank the reviewer for his/her enthusiastic comments that our manuscript is original and important. The manuscript have been revised accordingly as suggested.

1. Discuss the proposed strategy (benefits and limitations) considering the target pathology and the organs at risk.

- We thank the reviewer for the constructive suggestion. We followed the advice and add a section in the conclusion to discuss the benefits and limitations of our proposed combination drug delivery strategy with boron capsules. Please check as below:

“Application of this stable and biocompatible delivery system enables sustained drug release and enriched adequate boron to cancer cells for BNCT. As a local treatment technique, the abscopal effects of BNCT remain to be explored. Combinational therapy has shown efficacy to boost the abscopal effect both in preclinical and clinical studies. Of note, the boron neutron capture reaction can generate atomic defects or break the skeletons of two-dimensional materials, readily forming channels for drug transportation, thus empowers BNCT as a new function for controlled drug release. This application for the first time enabled us to treat tumours concurrently with BNCT and adjuvants, which uncovers an immunotherapy potential of BNCT.”

2. Discuss tumour selectivity in terms of boron carrier uptake.

- We thank the reviewer for the advice. As shown in **Figure R6**, notable intracellular uptake in both *in vitro* and *in vivo* experiments, suggesting boron capsules can be efficiently endocytosed by tumour cells. Nevertheless, we did find that boron capsules were also phagocytosed by macrophages, indicating that the boron capsules are not tumour-selective and that BNCT will kill all cells that have endocytosed boron capsules. As a local treatment technique, boron capsules are administered by intratumoural injection rather than intravenously, and tumour selectivity is unnecessary.

3. Describe/discuss the radiotoxic effects in normal tissues of the treatment described in the manuscript.

- As suggested, a comprehensive study has been performed to evaluate the safety issues of boron capsules after BNCT, including the routine blood test and Hematoxylin and Eosin (H&E) staining assay of hearts, livers, spleens, lungs and kidneys. Tumour-bearing mice treated with PBS, imiquimod, PEG-B-COF, and boron capsules, respectively, were sacrificed on day 14.

As shown in **Figure R4**, similar to the other experimental groups, the routine blood test of the boron capsule+N and PEG-B-COF+N group showed negligible cytotoxicity. In addition, all liver and kidney function parameters were within the normal range, reflected by the results of blood biochemical tests. Besides, as shown in **Figure R5**, no abnormalities were observed in the major organs of all experimental groups. Based on these experimental results, the boron capsule shows great biocompatibility and safety for the BNCT and holds the promise to potentiate the abscopal antitumour effect, which may be clinically practical to treat metastases through localized radiotherapy.

4. Describe the in vivo irradiations, features of the neutron source and neutron spectrum, animal shielding, irradiation dose components, absorbed dose delivered to tumour and to dose-limiting tissues... These data will allow the reader to interpret the data within the context of the vast existing literature on BNCT.

- As suggested, we have added a discussion of In-Hospital Neutron Irradiator (IHNI) in the revised manuscript.

“All experiments were performed at the In-Hospital Neutron Irradiator (IHNI) based on a miniature neutron source reactor, a low-power research reactor designed and manufactured by the China Institute of Atomic Energy (CIAE). The rated power of IHNI is 30 kW, the neutron flux is $1.9 \times 10^9 \text{ cm}^{-2}\cdot\text{s}^{-1}$ and the thermal neutron beam energy field of IHNI is $<0.4 \text{ eV}$.” (Ther Radiol Oncol 2018;2:49)

Besides, animal shielding has been comprehensively described in our recently published study². The fundamental characteristics of the neutron beams of IHNI-1, including the neutron spectrum, neutron fluence rate and its spatial distribution, as well as the dose induced by undesired neutrons and γ -rays of free neutron beams in-air, were experimentally determined by the research group from CIAE, and these studies have been published, and we have added the related references in the revised manuscript accordingly.

Dosimetric evaluation of irradiation were based on mean boron concentrations of tumour and normal tissues 24 h post administration. (**Table R1**)

	Thermal neutron dose rate (cGy/s)	Epithermal neutron dose rate (cGy/s)	Fast neutron dose rate (cGy/s)	Gamma dose rate (cGy/s)	B-10 dose rate (1 ppm) (cGy/s)	B-10 concentration (ppm)	Irradiation time (min)	B-10 dose (Gy)	Total dose (Gy)
Tumour tissue	0.0273	0.0006	0.0347	0.0151	0.0124	19.8	30	4.4030	5.8019
Normal tissue						0		0	1.3989

Table R1. Dose composition of normal and tumor tissues during neutron irradiation.

Reviewer #4

The manuscript presents the investigation “...of whether a carborane-based covalent organic framework (B-COF) can be based to develop a boron “capsule” of immune adjuvants for concurrent BNCT and immunotherapy”. To test their hypothesis the authors performed in vitro and in vivo experiments that are adequate but some critical issues must be addressed. The issues are listed below.

- We thank the reviewer for the constructive suggestions, and the manuscript have been revised accordingly.

1. To determine immune cell subtypes based on differentially expressed markers, which statistical test was used in Seurat for the differential expression analysis?

- Wilcoxon Rank Sum Test was used in Seurat for the differential expression analysis. As suggested, we have added the description in the revised manuscript.

2. Other concerns regarding the scRNA-seq portion of the study are the availability of the data (raw and Seurat processed) and the availability of the code used. Those are essential data and material to ensure that others can access and reproduce the results.

- As suggested, the related description has been corrected in the revised manuscript. The raw sequence data reported in this paper have been deposited in the Genome Sequence Archive (Genomics, Proteomics & Bioinformatics 2021) at National Genomics Data Center (Nucleic Acids Res 2021), China National Center for Bioinformation/Beijing Institute of Genomics, Chinese Academy of Sciences (GSA: CRA005110) that are publicly accessible at <https://ngdc.cncb.ac.cn/gsa/s/tyeq46vu> before Jun 2023.

3. The authors use bilateral tumour implantation in mice as a model of metastasis by treating one of the tumours while keeping the other side treatment intact to mimic the lack of irradiation in metastatic lesions in patients. However, in my opinion, this is not the most suitable murine metastatic cancer model. There are multiple protocols to induce metastasis in mice (e.g.: intravenous injection for lung colonization, one of the sites of colorectal cancer metastasis) and even models that develop spontaneous metastasis. Why did the bilateral model choose as a model of metastasis other than other models that better mimic metastatic disease and systemic irradiation effects? This is a critical experiment and the most adequate model impacts biological interpretation and relevance of the findings.

- As suggested, we have performed BNCT in a breast cancer model with lung metastases. The tumour was dissected 14 days after BNCT treatment, and statistical analysis of lung metastatic lesions was performed to evaluate the suppression efficacy of BNCT on lung metastases.

Figure R11. Metastasis inhibition in lung metastatic B16F10 and subcutaneous B16F10 tumour models by BNCT with immunological adjuvant-loaded boron capsule. The mice were treated with PBS, imiquimod, PEG-B-COF or boron capsule with neutron irradiation (n = 6). (A), Treatment scheme. (B), Representative lung photographs and H&E staining for each treatment group. Scale bar = 500 μ m. (C), Number of metastatic node per lung in different groups. Data shown as mean \pm s.d. (two-tailed unpaired Student's t-test, ****P < 0.001).

As shown in **Figure R11**, the degree of inhibition of lung metastasis was significantly higher in the PEG-B-COF+N group compared with the other groups. In addition, BNCT alone showed weaker inhibition of pulmonary metastases than the boron capsule + N group, suggesting that BNCT alone is difficult to elicit a potent immune response.

4. The comparisons between the bilateral tumours only show the cytokines and cell types proportions for the “primary tumours”. It is critical for biological interpretation that authors include for the “distant tumours” the results for serum cytokines and immune subtypes as shown for the “primary tumours” in Figures 6d-g. In the same way, the graphs demonstrating tumour growth curves for the “primary tumours” are essential.

- We apologize for the unclear presentation in the original submission. The proportion of immune cell types for “primary tumours” was shown in Supplementary Figure 20. The serum cytokines were analyzed in the serum of bilateral syngeneic B16F10 and MC38 mouse models, and the original tumour growth inhibition curve was present in Figure 4C-H.

5. Figure 1i refers to more than one panel. I recommend the authors label all panels, thus Figure 1i will be broken into i, j and k. Also, the scale for the bars is only shown in one panel and I recommend adding that to all.

- We thank the reviewer for this constructive advice. The figures have been corrected accordingly in the revised manuscript.

6. In Figure 4b, the scale bars are missing the size. The same happens to the other panels in Figure 4, where the size is only shown in the last image of each panel. Just to facilitate readers' comprehension, I recommend adding the size to all images.

- As suggested, the figures have been corrected accordingly in the revised manuscript.

7. Another issue with Figure 4 is the order of the panels. I recommend the authors place the panels in alphabetical order.

- As suggested, the figures have been corrected accordingly in the revised manuscript.

8. In Figure 5d, the proportions of myeloid, B and NK cells are shown grouped. I recommend breaking them to show each individual proportion instead of grouping the subtypes to facilitate results interpretation.

- As suggested, the figures have been replaced as **Figure R12** in the revised manuscript.

Figure R12. Relative tumour-infiltrating CD45⁺ immune cells frequencies of different clusters with and without BNCT.

Reference

- 1 Wang, Q. *et al.* A bioorthogonal system reveals antitumour immune function of pyroptosis. *Nature* **579**, 421-426 (2020).
- 2 Li, J. *et al.* Boron encapsulated in a liposome can be used for combinational neutron capture therapy. *Nature communications* **13**, 1-11 (2022).

REVIEWERS' COMMENTS

Reviewer #1 (Remarks to the Author):

I congratulate the authors again for the nice work and results. Most issues have been properly addressed. However, they have not been included in the final version of the manuscript. In the title it is clearly indicated that treatment "augments cancer immunotherapy". Thus, data related to mechanism of cell death and immunomodulation in the tumor microenvironment are relevant, as well as, systemic immunotoxicity + pharmacokinetics.

I do not recommend the publication of the manuscript without including the following data:

- pharmacokinetics - systemic toxicity: Figures R4 and R5 must be included in supp. info.
- New data related to cell death and uptake are relevant: Figures R1 and R2 must be included in supp. info.
- Activity of treatment on immune cells is key for the final antitumoral activity. Thus, I insist on including these data (Figure R3 (B-H)) in the manuscript (not as supp. info.).
- Metastasis inhibition in Figure R11 is also relevant and should be added to the manuscript.
- I cannot find revised Figure R12 in the revised manuscript.

Reviewer #2 (Remarks to the Author):

The authors have appropriately responded to majority of the questions. There are still some minor concerns.

1. As for the first concern in the previous comments, we were puzzled about the penetration of boron capsules in tumor tissues, because the particle size of boron capsules is too large, which will cause diffusion difficulties after intra-tumoral injection.
2. In the Figure 1, schematic representation of synthesis of B-COF, where B CHO is used as a linker, and the reaction between 1,3,5-tris(4-aminophenyl)-benzene (TAPB) and p-carborane-1,10-phenyl-dialdehyde (B CHO), will it produce other products? Such as dendrimers (a highly branched, symmetrical and radial type of functional polymer) or hyperbranched Polymers.
3. Considering natural abundance of ^{10}B is only about 20%, does this mean that the B-COF disintegrates incompletely after radiation exposure (Fig. 1B)? which may result in incomplete release of drugs.

Reviewer #3 (Remarks to the Author):

The authors have addressed the issues I raised in my review carefully and comprehensively.

However, I would like them to address 2 minor points:

In my previous review I asked the authors to discuss the benefits/limitations of the treatment strategy bearing in mind one or more tumor targets and the corresponding dose-limiting tissues. They did not address this issue. I think it is useful to understand which tumor entity/entities the authors consider this strategy might be most useful for.

Also, in the response letter, the irradiation dose table is called Table R1 when in fact it is part of the Supplementary material and is called Table S11. If possible, I would move this table to the main text.

Reviewer #4 (Remarks to the Author):

I value the efforts of the authors to address all comments from all reviewers which included the execution of additional experiments that I consider were properly designed to tackle the issues raised. Regarding specifically my comments and critics, I am satisfied with the responses and appreciate the metastatic mouse model that was included to examine the efficacy of the boron capsule in reducing metastasis development. That was performed in addition to the bilateral model that to my knowledge was not as robust as the breast cancer model that develops lung metastasis that was included after the initial review of the manuscript. With this new model, the authors confirm the results previously shown.

The authors also addressed the data sharing by depositing the single-cell RNA-seq at GSA (Genome Sequence Archive). However, one issue remained unaddressed that is the sharing of the code used for the single-cell analysis. To ensure that other scientists can reproduce the single-cell results observed by the authors, sharing the code is crucial and should be done prior to the publication of the manuscript.

In summary, I am satisfied with the authors' response to the initial review and recommend the manuscript for publication. Though, I highly recommend that the code is shared with the publication of the manuscript.

RESPONSES TO THE COMMENTS

Dear reviewers,

On behalf of the authorship group, I would like to thank the reviewers for taking the time to evaluate our manuscript entitled “**Localized Nuclear Reaction Breaks Boron Drug Capsules and Augments Cancer Immunotherapy**” (NCOMMS-22-09499)”. We are deeply grateful for your appreciation on our work. Please kindly find our point-by-point response to reviewers’ comments below.

Reviewer #1

I congratulate the authors again for the nice work and results. Most issues have been properly addressed. However, they have not been included in the final version of the manuscript. In the title it is clearly indicated that treatment "augments cancer immunotherapy". Thus, data related to mechanism of cell death and immunomodulation in the tumor microenvironment are relevant, as well as, systemic immunotoxicity + pharmacokinetics.

We thank the reviewer for his/her encouraging comments and constructive advice. The manuscript and Supplementary Information have been revised accordingly.

1. Pharmacokinetics - systemic toxicity: Figures R4 and R5 must be included in supp. info
- As suggested, we have moved the corresponding Figure to the revised Supplementary Information named as Supplementary Figure 18 and S19
2. New data related to cell death and uptake are relevant: Figures R1 and R2 must be included in supp. info.
- Thanks for your suggestion, the mentioned Figures are included in the revised Supplementary Information named Supplementary Figure 14 and S25
3. Activity of treatment on immune cells is key for the final antitumoral activity. Thus, I insist on including these data (Figure R3 (B-H)) in the manuscript (not as supp. info.).
- Thank you for your constructive suggestion about our description of cancer treatment. We have removed the relevant data to Figure 5G-K in the revised manuscript. However, due to the limited space in Figure 5, we have to put the data for the population of CD3+ and M2 macrophages in Supplementary Fig. 24A-B.
4. Metastasis inhibition in Figure R11 is also relevant and should be added to the manuscript.
- As suggested, the treatment result of the metastasis tumour is now shown in Figure 6L-M and Supplementary Figure 27 in the revised manuscript.
5. I cannot find revised Figure R12 in the revised manuscript.
- We apologize for the missing of Figure R12 and we have added it to Figure 5D in the revised manuscript.

Reviewer #2

The authors have appropriately responded to majority of the questions. There are still some minor concerns.

- We thank the reviewer for the constructive suggestions, and the manuscript has been revised accordingly.

1. we were puzzled about the penetration of boron capsules in tumor tissues, because the particle size of boron capsules is too large, which will cause diffusion difficulties after intra-tumoral injection.

- We agree with the reviewer that large-size particles will face the problem of diffusion. In general, we perform multi-directions injection around the tumor during the intratumoural administration to make sure the boron capsule is distributed as evenly as possible in the tumor. Besides, the anti-tumour effect of boron capsule is owing to both BNCT and the subsequent activation of the antitumoural cytotoxic T-cell, which remodels the tumour immune microenvironment. Therefore, the diffusion deficiency of the boron capsule does not significantly influence its anti-tumour activity.

2. In the Figure 1, schematic representation of synthesis of B-COF, where B CHO is used as a linker, and the reaction between 1,3,5-tris(4-aminophenyl)-benzene (TAPB) and p-carborane-1,10-phenyl-dialdehyde (B CHO), will it produce other products? Such as dendrimers (a highly branched, symmetrical and radial type of functional polymer) or hyperbranched Polymers.

- Generally speaking, a certain percentage of disordered oligomers would be produced during the preparation of covalent organic frameworks including B-COF, but they usually dissolve well in organic solvents. Thus, in the case of the preparation of B-COF, the crude product was washed with acetonitrile and methanol three times to remove the potential by-products. Consistently, the results of PXRD and TEM indicated the crystal structure of B-COF was highly ordered (**Figure R1**), which is hardly reported in the field of dendrimers.

Figure R1. (A), Powder X-ray diffraction (PXRD) patterns of the experimentally observed (red curve), Pawley refined (grey curve) B-COF and an eclipsed AA stacking mode (blue curve) (their difference is shown in the green curve). (B, C) transmission electron microscopy (TEM) of B-COF with uniform morphology.

3. Considering natural abundance of ^{10}B is only about 20%, does this mean that the B-COF disintegrates incompletely after radiation exposure (Fig. 1B)? which may result in incomplete release of drugs.

- We understand the concern about the incomplete release of drugs from the reviewer. However, the release of drugs is owing to the vacancies caused by the energy deposit of ^4He and ^7Li , of which energies

were 2-3 magnitudes higher than the energy of chemical bonds. Instead of generating only one atom defect of ^{10}B , the high energy recoil particles (^4He and ^7Li) can knock out many atoms, which may continue striking other atoms in different layers, leading to the generation of a large vacancy (**Figure R2 and Attachment File 1-2**). In the process of drug release, the neutron capture reaction of ^{10}B is more of a switch rather than the generator of defects. Thus, with these defects, drugs could be released in a sustained manner over a month *in vitro*.

Figure R2. Schematic illustration of defect formations investigated by molecular dynamics (MD) simulation with a simplified graphene model. Step 1, A high flux thermal neutron reacted with carborane through $^{10}\text{B}(n,\alpha)^7\text{Li}$ reaction; Step 2, The fission-generated high-energy ions (i.e. alpha particles and lithium-7 nuclei) deposits energy through inelastic collisions to atoms in B-COF; Step 3, The high-energy ions break B-COF framework and produce many secondary ions, which further promote the development of defects

Reviewer #3

The authors have addressed the issues I raised in my review carefully and comprehensively. However, I would like them to address 2 minor points:

1. Discuss the benefits/limitations of the treatment strategy bearing in mind one or more tumor targets and the corresponding dose-limiting tissues.

- Local treatment technique is now of highly clinical application. However, owing to the poor penetration, the effectiveness of local administration is not as potent as systemic administration in the treatment of multiple tumours. In the present work, we attempted to address this problem to a certain extent by combining BNCT with an immune agonist strategy by drug transportation, which is achieved through the atomic defects or the breaking of B-COF structure generated by the boron neutron capture reaction and proved this strategy can be utilized in the treatment of primary tumour, distant tumour and metastasis.

2. If possible, please move Table S11 to the main text.

- As suggested, Table S11 is moved to the main text as Figure 4C in the revised manuscript.

Reviewer #4

I value the efforts of the authors to address all comments from all reviewers which included the execution of additional experiments that I consider were properly designed to tackle the issues raised.

Regarding specifically my comments and critics, I am satisfied with the responses and appreciate the metastatic mouse model that was included to examine the efficacy of the boron capsule in reducing metastasis development. That was performed in addition to the bilateral model that to my knowledge was not as robust as the breast cancer model that develops lung metastasis that was included after the initial review of the manuscript. With this new model, the authors confirm the results previously shown.

- Thanks for the reviewer for his/her encouraging comments on our work. The treatment result of the lung metastasis model is shown in Figure 6L-M in the revised manuscript.

1. However, one issue remained unaddressed that is the sharing of the code used for the single-cell analysis. To ensure that other scientists can reproduce the single-cell results observed by the authors, sharing the code is crucial and should be done prior to the publication of the manuscript.

- Thank you for your constructive suggestion. However, we performed the single-cell sequencing in collaboration with Abioscience, Inc. and due to the company's commercial confidentiality, we are afraid that we could not share the full code. If needed, readers can contact with the company for more detail.